# Trend Analysis of Temperature Data for the Narayani River Basin, Nepal

Mohan Bahadur Chand [1,*], Bikas Chandra Bhattarai [2], Niraj Shankar Pradhananga [3] and Prashant Baral [4]

1 Graduate School of Environmental Science, Hokkaido University, Hokkaido 060-0810, Japan
2 Department of Geosciences, University of Oslo, Blindern, 0316 Oslo, Norway; b.c.bhattarai@geo.uio.no
3 Department of Hydrology and Meteorology, Government of Nepal, Kathmandu 44600, Nepal; nirajsp@hotmail.com
4 Geographic Information Systems (GIS) Area, NIIT University, Rajasthan 301705, India; prashant.baral@st.niituniversity.in
* Correspondence: mohanchand@ees.hokudai.ac.jp

**Abstract:** The study of spatiotemporal variation in temperature is vital to assess changes in climate, especially in the Himalayan region, where the livelihoods of billions of people living downstream depends on water coming from the melting of snow and glacier ice. To this end, temperature trend analysis is carried out in the Narayani River basin, a major river basin of Nepal, characterized by three climatic regions: tropical, subtropical and alpine. Temperature data from six stations located within the basin were analyzed. The elevation of these stations ranges from 460 to 3800 m a.s.l. and the time period of available temperature data ranges from 1960–2015. Multiple regression and empirical mode decomposition (EMD) methods were applied to fill in missing data and to detect trends. Annual as well as seasonal trends were analyzed and a Mann–Kendall test was employed to test the statistical significance of detected trends. The results indicate significant cooling trends before 1970s, and warming trends after 1970s in the majority of the stations. The warming trends range from 0.028 to 0.035 $^\circ$C year$^{-1}$ with a mean increasing trend of 0.03 $^\circ$C year$^{-1}$ after 1971. Seasonal trends show the highest warming trends in the monsoon season, followed by winter and the premonsoon and postmonsoon season. However, the difference in warming rates between different seasons was not significant. An average temperature lapse rate of $-0.006$ $^\circ$C m$^{-1}$ with the steepest value ($-0.0064$ $^\circ$C m$^{-1}$) in the premonsoon season and the least negative ($-0.0052$ $^\circ$C m$^{-1}$) in the winter season was observed for this basin. A comparative analysis of the gap-filled data with freely available global climate dataset show reasonable correlation, thus confirming the suitability of the gap filling methods.

**Keywords:** climate change; temperature trend; Himalaya; river basin; Nepal

## 1. Introduction

Global warming and climate change are widely recognized as the most significant dilemmas the world is experiencing today [1]. Studies based on direct measurements and remote sensing have suggested that higher greenhouse gases in the atmosphere are causing global climate change [2]. Global temperatures have been rising significantly over the last decade, despite year-to-year fluctuations associated with the El Niño–La Niña cycle of tropical ocean temperature [3]. Linear warming trends from 1951 to 2012 show an increase in global temperature of 0.12 $^\circ$C decade$^{-1}$ [2]. However, the rate of temperature increase is different for different regions. The widespread retreat of glaciers and snow cover due to increasing temperatures has contributed to eustatic sea level rise [2]. This warming will also have significant impacts on the hydrological cycle of mountain river basins, affecting the livelihoods of populations living downstream.

The Hindu-Kush Himalayan (HKH) region is characterized by mountainous environments [4]. Weather and climatic conditions over the Himalayan regions are of great

interest to the scientific community [5]. HKH has the highest elevation range in the world and contains largest freshwater reserve in the form of snow and glacier ice outside the polar regions. Several studies have reported warming in the Himalayan region [4,6–10] which has caused shrinkage of glaciers [11–13], expansions of glacier-fed lakes [14–17], degradation of permafrost [18] and changes in the hydrological cycle of many mountain rivers [19,20]. For example, Lama et al. [13] reported a loss of about 22% of the glacier area in Mustang, within the Narayani basin from the 1980s–2010s. The temperature is increasing faster than the global average in the Tibetan Plateau and Northern Hemisphere [10], and the Himalayan region, and the influence of rising temperatures is greater in the Eastern Himalayas compared to that in the greater Himalayas [21]. Shrestha et al. [7] has analyzed the maximum temperature since 1977 and found a warming trend of temperature of 0.06 to 0.12 $°C$ $year^{-1}$ in the Middle Mountain and Himalayan region, and of 0.03 $°C$ in Siwalik and the southern plains. Kattel and Yao [9] reported an increase in temperature of 0.038 $°C$ $year^{-1}$ in mountain stations of Nepal Himalaya. Khatiwada et al. [8] found that increases in maximum temperature (0.05 $°C$ $year^{-1}$) were larger than those of the minimum temperature increase (0.01 $°C$) from 1981–2012, with the largest change observed during the premonsoon season in the Karnali River basin. Similarly, Nepal [22] studied the impact of climate change on the hydrological regime of the Koshi river basin and found an increase in the average maximum temperature of 0.058 $°C$ $year^{-1}$ and 0.014 $°C$ $year^{-1}$ for minimum temperature over the past 40 years. The model result of Nepal [22] also shows that the annual discharge of the Koshi River basin will increase by 13% by the mid-century and snowfall will decrease substantially due to increasing temperature. A more recent study by Shrestha et al. [23] also showed an increasing trend of seasonal maximum and minimum temperature from 1975–2010 in the Koshi trans-boundary basin. Similar study by Yang et al. [5] shows that air temperature has increased by 0.62 $°C$ $decade^{-1}$ over the last 49 years (1959–2007) at Dingri station, northern slope of Mt. Everest. Liu et al. [24] also observed a cooling trend of about $-0.06$ $°C$ $decade^{-1}$ in three stations out of 88 stations studied in the Tibetan Plateau. Seasonally, some contrasting results are reported by previous studies. Duan and Xiao [10] reported the highest warming trend in the monsoon season in Tibet. Liu et al. [24] and Yang et al. [5] found higher increasing trends in winter that ranged from 0.36 to 0.86 $°C$ $decade^{-1}$ in the same region. Similarly, Shrestha et al. [7] also reported highest increasing rate during the winter season and lowest in premonsoon season in whole Nepal. Few past studies also showed a relation between temperature and elevation to estimate the temperature-elevation gradient and found 0.72 ± 0.01 $°C$ 100 $m^{-1}$ [5] in Tibet and 0.50 $°C$ 100 $m^{-1}$ in Langtang Valley, Nepal [25].

HKH regions have experienced an overall rapid warming, which has further influenced the climatic extremes and hydrological cycles in the region [26]. Several studies related to temperature trends have been carried out in this region using observed, reanalysis and satellite data sets. However, river basin scale studies of temperature trends are fairly limited in Nepal Himalayas. This study chooses the Narayani River basin to study the mean annual and seasonal temperature trend.

## 2. Study Area

The Narayani River basin (Figure 1), also known as the Gandaki River basin, lies in the central part of Nepal between 25.49–29.28° N and 85.02–85.83° E. The Narayani River basin is a transboundary river basin that originates from the southern edge of the Tibet (China), flows through Nepal to India, and contributing to the Ganges River. It encompasses a total area of 46,300 $km^2$ [11], 32,104 $km^2$ [27,28] of which is in Nepal. The Narayani River has seven tributaries, i.e., Marsyandi, Daraudi, Seti, Madi, Kali Gandaki, Budhi Gandaki, and Trishuli, and it includes Terai, Hill, Mountain and High Himalayas, including trans-Himalaya from south to north of the basin. About 40% of the basin is covered by agricultural land [27]. The upper part of this basin is covered with snow and ice, which is the main source of water. The mean elevation of the Narayani River basin is 4065 m a.s.l. and ranges from 180 m in the south to higher than 8000 m [29] in the north, where the basin contains

the Dhaulagiri (8167 m a.s.l.), Manaslu (>8000 m a.s.l.) and Annapurna (8091 m a.s.l.) peaks, which have very contrasting climates. Langtang, Machhapuchhre, Manaslu, Dhawalagiri and Annapurna mountains in the basin have created the large variation in climate over the basin.

The Narayani River basin is dominated by monsoons, with annual average precipitation ranges from 152 to 5493 mm. About 78% of annual precipitation occurs during the monsoons (June–September) [28,30]. The lowest precipitation is observed for the driest regions, including Mustang and Manang, located at the leeward-side north of the Annapurna, and Lumle receives the highest precipitation [31].

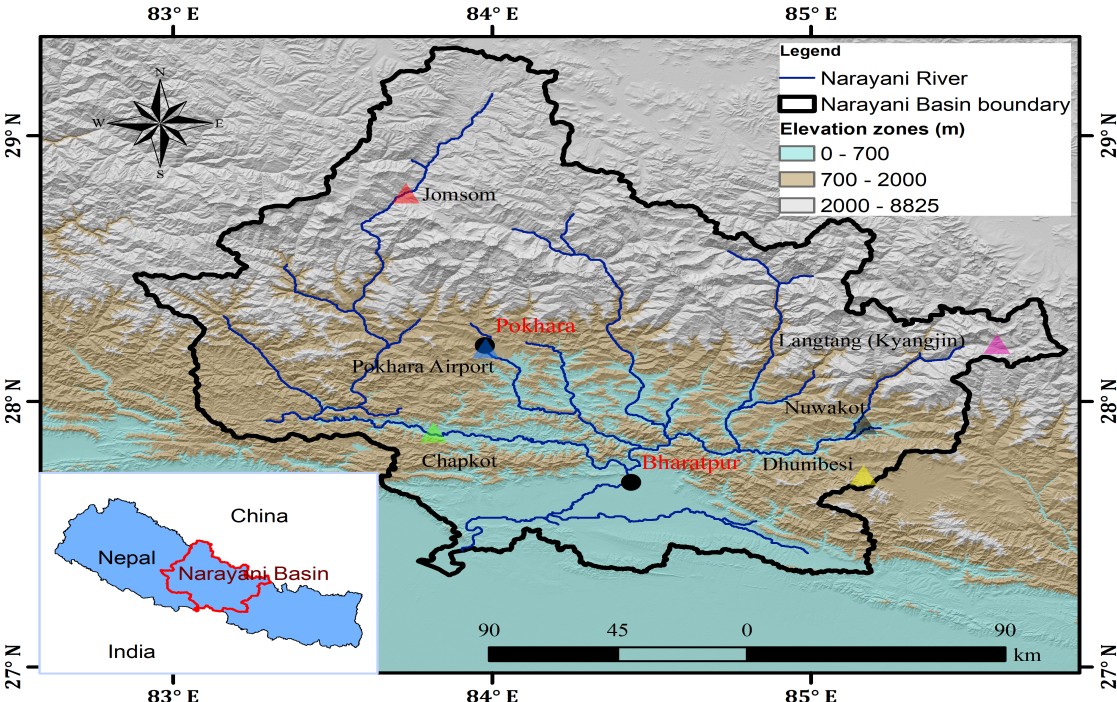

**Figure 1.** The Narayani River basin with the location of the meteorological stations used in the study. The background of the basin shows different elevation zones, based on the Shuttle Radar Topography Mission Digital Elevation Model.

## 3. Methods

### 3.1. Data

Daily two meter temperature data of The Narayani River basin were obtained from the Department of Hydrology and Meteorology (DHM), Government of Nepal. Six stations were selected (Table 1), each representing different physiographic zones and elevation ranges from 460 (Chapkot) to 3800 m a.s.l. (Langtang). The data range from 1 January 1956 to 31 December 2016. None of the stations has data for the entire period (Figure 2) and some of them have data gaps longer than 5 years. Langtang station has the shortest temperature record (1988–2008) and Nuwakot has the longest record (1956–2016) with significant gaps in between. Data available for selected stations were of different observation periods and missing observation dates. Missing data in each station were interpolated by linear and multiple regression methods (described in Section 3.3).

### 3.2. Basic Statistics

Langtang (Station ID 1000) is the highest elevation station among six stations (Figure 3), where a daily mean temperature of 3.19 °C (Table 2) was observed. Nuwakot (Station ID: 1004) has the longest period (about 60 years) of observation, from 1956 to 2015, with a daily mean temperature of 21.53 °C. Table 2 also shows the 95th and 99th percentile of recorded temperature and makes it clear that data in all stations skewed towards the right and were not normally distributed. The similar magnitude of skewness coefficient indicates similar

seasonality in all stations. Auto-correlation for each station data was calculated and found more then 0.8 for one-day time lag.

**Table 1.** Geographic coordinates of the six weather stations.

| Station ID | Station Name | Lat. | Lon. | Altitude (m) |
|---|---|---|---|---|
| 810 | Chapkot | 27.883 | 83.817 | 460 |
| 804 | Pokhara Airport | 28.217 | 84.000 | 827 |
| 1004 | Nuwakot | 27.917 | 85.017 | 1003 |
| 1038 | Dhunibesi | 27.171 | 85.183 | 1085 |
| 601 | Jomsom | 27.783 | 83.717 | 2744 |
| 1000 | Langtang | 28.200 | 85.533 | 3800 |

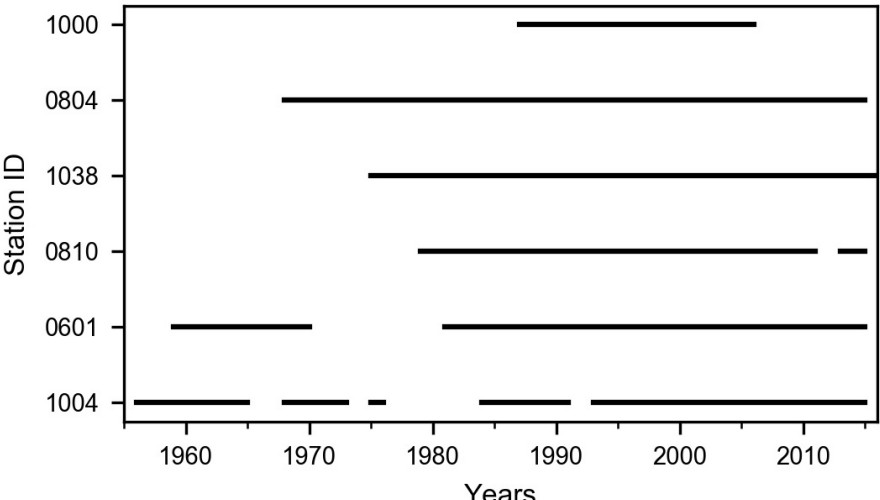

**Figure 2.** Data coverage for the studied stations. Gaps in the lines indicate no available data for the period.

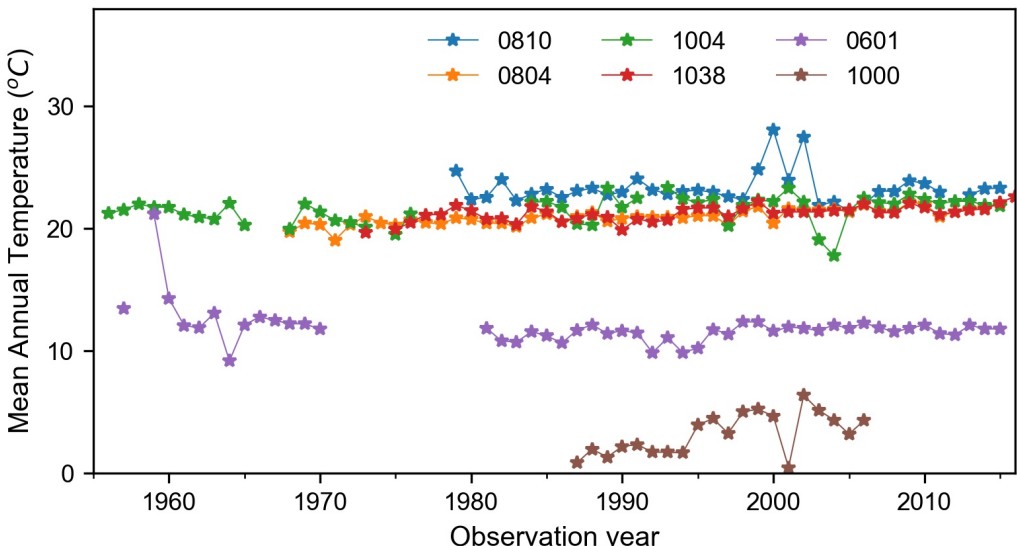

**Figure 3.** Time series of observed mean annual temperature (°C) without filling the gap.

### 3.3. Quality Control and Data Fill

All available data-sets for six stations from 1960–2015 were visually inspected using several time series plots and outliers, $+/-$ three standard deviations [32], were removed. Simple and multiple linear regression methods with ordinary least squares (OLS) assumptions were applied to fill the data gaps. Independent variables for the model were selected based on the highest significant correlation coefficient with surrounding stations in the model development process to fill the gap. To improve the predictive power of the model, step-wise regression methods were used to select multiple independent variables. Only overlapping dates were used while developing the models and used for filling data gaps in studied stations.

**Table 2.** Distribution of temperature data over six observation stations.

| Station ID | 95th Percentile | 99th Percentile | CV | Mean | Median | Skew | Mean after Interpolation |
|---|---|---|---|---|---|---|---|
| 810 | 28.85 | 31.68 | 26.36 | 23.27 | 24.77 | −0.59 | 23.20 |
| 804 | 26.38 | 26.85 | 21.66 | 20.96 | 22.38 | −0.46 | 21.06 |
| 1004 | 26.51 | 27.88 | 20.69 | 21.53 | 22.86 | −0.59 | 21.45 |
| 1038 | 26.73 | 27.70 | 23.71 | 21.20 | 22.59 | −0.50 | 21.19 |
| 601 | 18.85 | 20.77 | 29.53 | 11.77 | 11.68 | −0.08 | 11.44 |
| 1000 | 8.20 | 9.13 | 11.29 | 3.19 | 3.36 | −0.20 | 3.16 |

Step-wise regression is a semi-automated process of building a model by successively adding or removing variables based solely on the t-statistics of their estimated coefficients [33]. The step-wise regression options have more power and information than the ordinary multiple regression option and is especially useful for shifting through large numbers of potential independent variables and/or fine-tuning a model by poking variables in or out. When improperly used, it may converge on a poor model while giving a false sense of security. The step-wise option lets us either begin with no variables in the model and proceed forward (adding one variable at a time) or start with all potential variables in the model and proceed backward (removing one variable at a time). At each step, it computes different statistics and selects the best independent variables, based on F-test. Although the selected stations are in different climatic zones when developing the regression models to fill gaps, independent variables selection was solely based on statistically significant ($p < 0.05$) correlations with independent variables so the impact of physiography and elevation were assumed to be minimal.

High R-squared values ($>0.80$) were observed with less scatter of observed and predicted mean annual temperature around the best fit (Figure 4) when dataset from Jomsom, Dhunibesi, and Chapkot stations were used as dependent variables and Pokhara was used as the independent variable. While comparatively smaller value ($R^2 = 0.66$) was obtained when data from Langtang station were used as the dependent variable and data from Pokhara and Jomsom stations as dependent variables. The mean annual temperature after filling the gaps (Figure 5) was slightly lower ($\tilde{0}.10$ °C) than the mean annual temperature (Table 2) before filling the gaps.

### 3.4. Trend Break Detection Methods

Empirical mode decomposition (EMD) method [34,35], which is an adaptive method that is entirely empirical and captures the characteristics, was used for detecting the trend in different periods. Mhamdi et al. [35] have suggested that the EMD can be considered as an attractive and easy method for time series analysis and especially for time series trend extraction and, therefore, this method was adopted in this study. The key idea of EMD is to locally decompose data into oscillatory components—so-called intrinsic mode functions (IMFs). In this study, we used the EMD method for trend break detection in all six stations; however, the only result of the longest time-series (Pokhara station) is presented for brevity.

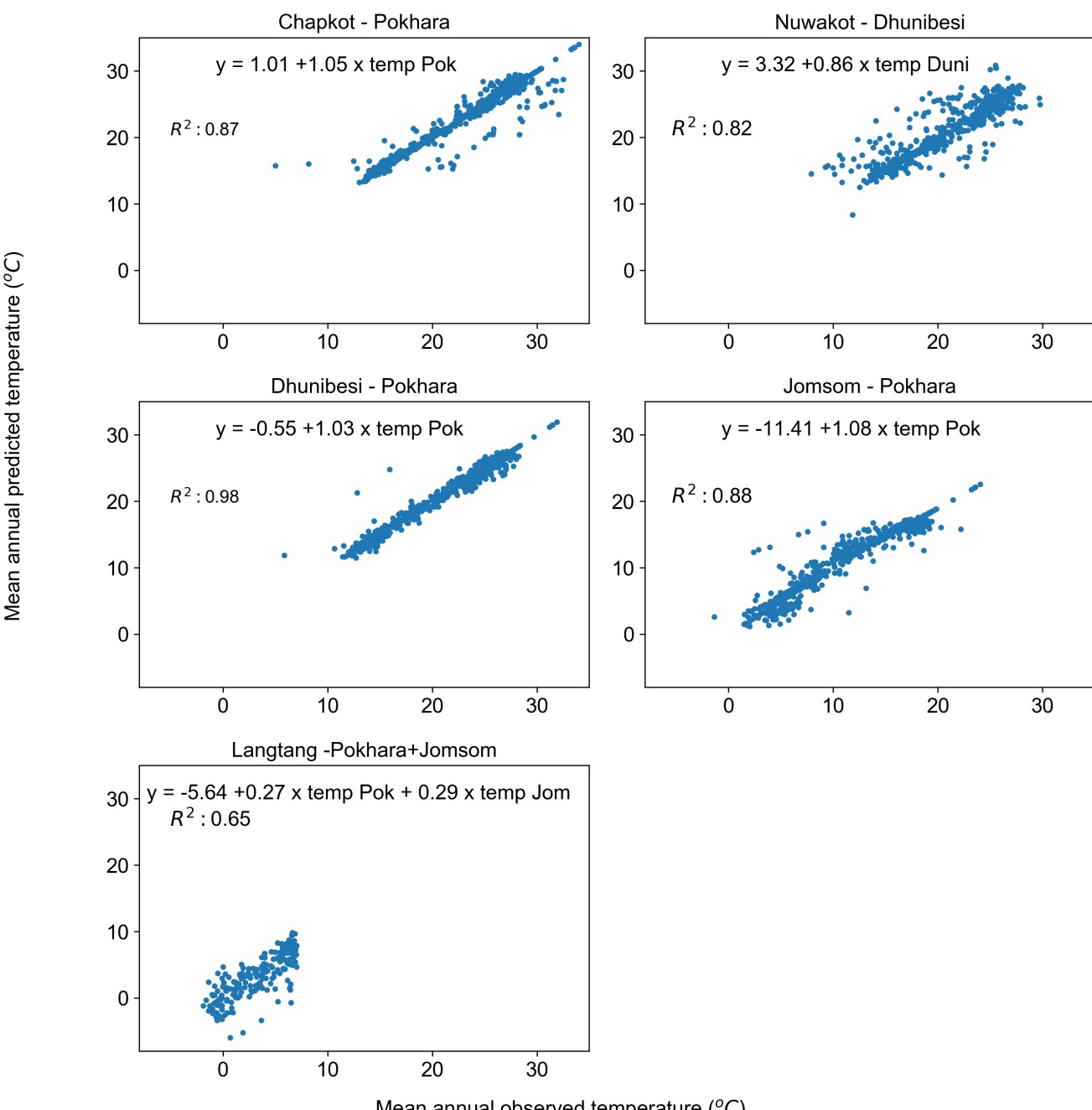

**Figure 4.** Scatter plot of temperature observations with the prediction from the model for the different stations.

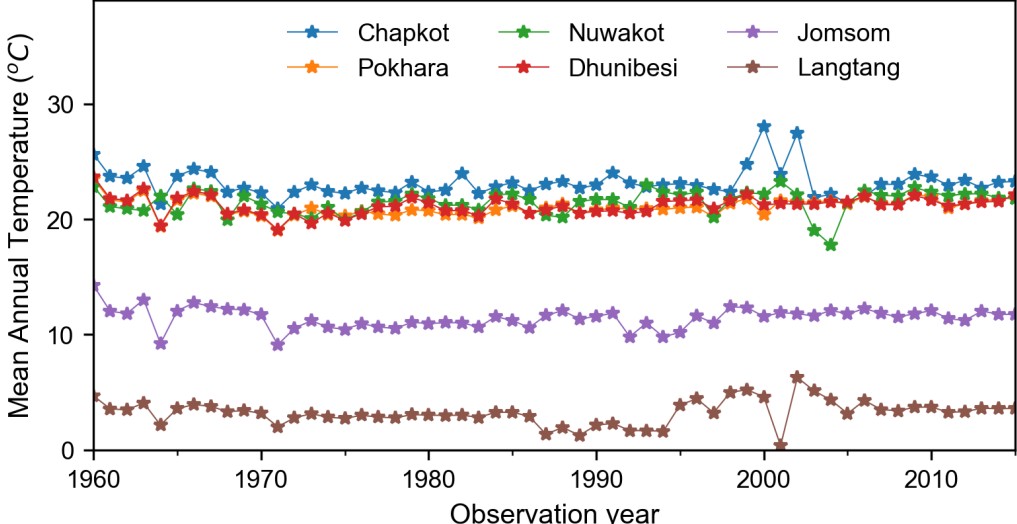

**Figure 5.** Time series plot for annual mean temperature after filling the missing data.

### 3.5. Seasonal and Annual Trend Analysis

The Mann–Kendall Test is a widely used method by several researchers for long-term climatic variable trend analysis, e.g., Karki et al. [31], Khatiwada et al. [8], Panthi et al. [28], Razvai et al. [36], Shrestha et al. [7] and Xu et al. [37]. Trend analysis was performed using Mann–Kendall non-parametric statistical test on the annual long-term trend for the historical period (1960–2015) of temperature time series. Seasonal trends were studied using historical data from 1970–2015 as trend break was observed in the 1970s for most of the stations. The Mann–Kendall (MK) test [38,39] statistically assesses if monotonic upward or downward trends are statistically significant or not. A monotonic trend means that the variable consistently increases (or decreases) through time, but the trend may or may not be linear. Basically, the Mann–Kendall test can be used in place of a parametric linear regression analysis, which can be used to test if the slope of the estimated linear regression line is different from zero. The regression analysis requires that the residuals from the fitted regression line should be normally distributed; an assumption not required by the MK test; that is, the MK test is a non-parametric (distribution-free) test [38]. However, there were some key assumptions associated with this test, such as the requirement for observations obtained over time that need to be independent and identically distributed, meaning a very long record and non-stationarity of the time series [39,40]. Hirsch et al. [41] suggested that the MK test can be best viewed as an exploratory analysis and is most appropriate to identify stations where changes were significant or of large magnitude and to quantify these findings.

### 3.6. Comparison of Station Data with Freely Available Global Climate Dataset

The gap filled daily mean temperature data from six stations were compared with global climate dataset from freely available sources. For this purpose, WorldClim Version 2.0, CHELSA (Climatologies at high resolution for the earth's land surface areas) Version 1.2, GLDAS (Global Land Data Assimilation System) Version 2.0 and MODIS (Moderate Resolution Imaging Spectroradiometer) satellite dataset were used.

WorldClim Version 2.0 comprises spatially interpolated and gridded climate data for global land areas at a very high spatial resolution (approximately 1 km$^2$), which are based on interpolation methods applied to meteorological station data [42]. The average monthly temperature data for 1970–2000 and "Annual Mean Temperature or BIO1 grid" from WorldClim Version 2.0 were used in this study. CHELSA Version 1.2 is a high resolution (30 arc sec) climate data set for the earth land-surface areas based on a quasi-mechanistical statistical downscaling of the ERA interim global circulation model [43].

The average monthly temperature data for 1979–2013 and "Annual Mean Temperature" grid from CHELSA Version 1.2 were used in this study. GLDAS Version 2.0 uses satellite data products, ground-based observational data products, land surface modeling and data assimilation methods to generate global climate surfaces [44,45]. The parameter "near surface air temperature" in Kelvin from GLDAS Noah Land Surface Model L4 monthly 0.25 × 0.25 degree Version 2.0 data sets were used in this study.

MOD11A2 Version 6.0 product, one of the several MODIS satellite products, was used in this study [46,47]. MOD11A2 Version 6.0 provides the average 8 day per-pixel land surface temperature (LST) in a 1200 × 1200 kilometer grid, where each pixel value has a spatial resolution of one kilometer and the pixel value is a simple average of all the corresponding MOD11A1 LST pixels collected within that 8-day period. LST daytime and LST night-time were extracted from the MOD11A2 grid for all six stations and separately averaged to obtain monthly average LST daytime and monthly average LST night-time. Monthly averages of LST daytime and LST night-time were further averaged to obtain monthly average LST for all months at all six stations. Since a single grid represents the 8-day average, a grid which is averaged using 8 days from two adjacent months was placed in the month with larger number of days used to create the 8-day average grid. Further, when a grid was averaged using 4 equal days in two adjacent months, the grid was placed in both months to create the monthly average.

As data from these sources are freely available in the form of grids, these grids were uploaded as raster data in ArcMap 10.5. The grids and station coordinates were projected into the same geographic projection using the WGS-84 datum and the function "Extract Multi Values to Points" in ArcMap was used to extract the pixel values at station coordinates. The Raster Calculator function in ArcMap 10.5 was used to convert digital number values of the MOD11A2 grid into degrees Celsius and also to convert temperatures in Kelvin in the GLDAS 2.0 data set to degrees Celsius.

For comparison with WorldClim 2.0 monthly averages and annual mean temperatures, station data from 1970 to 2000 were used to generate monthly averages and mean annual air temperature (MAAT hereafter). Similarly, for comparison with CHELSA Version 1.2 monthly averages and annual mean temperatures, station data from 1979 to 2013 were used to generate monthly averages and MAAT. For comparison with the GLDAS Version 2.0 monthly averages, the GLDAS Version 2.0 monthly temperature data set ranging from 2001–2010, and station data from 2001 to 2010, were used to generate monthly averages and MAAT. For the comparison of the MOD11A2 Version 6.0 product with station data, MOD11A2 Version 6.0 for the year 2014 and 2015 was compared with station data for the years 2014 and 2015, respectively. Further, the GLDAS Version 2.0 monthly temperature data set for the year 2014 and 2015 was also used for comparison.

Scatter plots were created and linear trend lines were fitted to observe the strength of linear correlation between station dataset and the freely available climate dataset at all stations.

## 4. Results and Discussion

### 4.1. Trend Break Observation

EMD methods were used to detect temperature trends in six stations within the Narayani River basin. Intrinsic mode function (IMF) 1 exhibits high frequency and can represent very short-term fluctuation (Figure 6), IMFs 2–3 capture a small percentage of variance, IMFs 4–5 capture mid-term effects described by periodic cyclic variation [48]. Finally, IMF 6, the residue component in EMD, could represent the major trend [49] of annual average temperature in the long term that may be related with the increase or decrease in observed temperature in Pokhara station. From Figure 6, high fluctuation during the periods between 1960 and 1970 is observed in IMF1. Higher fluctuation represents higher temperature variability during the period 1960 and 1970. Higher variability during that periods impacts mid periodic cycle, shown by IMF4, as similar results are also discussed by [48,49] in their research. The annual average temperature is in a decreasing trend

until 1972 and it breaks there, and again the trend is in increasing order and represents the major trend of temperature (Figure 6). This may be related to rapid global warming after the 1970s with industrial evolution and an increase in greenhouse gases (GHGs) and radiative forcings [2]. The small difference in the number of years of trend breaking might be attributed to the influence of local and regional climate.

### 4.2. Historical Observed Trend

The observed historical mean annual temperature anomalies after filling the gaps for the period of 1960–2015 are shown in Figure 5. This data set represents the basis for calculation of the annual and seasonal trend for each station. Few extreme events of temperature anomalies ($\pm 2$ °C), relative to the long-term mean, were observed in the data and also show the spatial variability in temperature, as stations are located in different physiographic zones. The average annual temperature of the basin is 16.92 °C; however, the temperature varies largely with altitude.

Trends of mean annual temperature for six stations are shown in Figure 7. Analyses of mean temperature data from six stations within the Narayani basin for the period 1960–2015 reveal a warming trend usually after 1970s and a cooling trend before this period. A trend break was observed in the 1970s for all stations, i.e., in 1970 for Jomsom and Pokhara, in 1972 for Chapkot and Dhunibesi, in 1979 for Nuwakot, which is cross-validated with the EMD results. However, the cooling trend for Langtang was detected in 1993, in which the original dataset ranged from 1988–2008 and data were extended for 1960–2015 by gap-filling techniques. The earlier period was dominated by the cooling trend in most of the stations and it might have influenced the data-filling method, and the cooling trend for Langtang was extended up to 1993 from where continuous data were available. Overall, all six stations show the warming trends (Table 3) and we observed negative trends in the earlier periods. Studies related with temperature trends in the Narayani River basin are not well documented. Here, we used the dataset from much earlier than the previous study, which were conducted either on a regional scale or in different river basins. Our results of warming rates from the Narayani basin are similar to Shrestha et al. [7], which was done for the whole of Nepal. The majority of stations (four out of six) in a later period, and three stations out of six stations for the entire period, show increasing trends of mean annual temperature at a statistically significant rate. The warming trends range from 0.008 to 0.015 °C with a mean warming trend of 0.011 °C year$^{-1}$ which is statistically significant at the 95% significance level (Table 3) for the entire period (1960–2015). However, analyses from a later period that starts in 1971 reveal higher warming trends, which range from 0.028 to 0.035 °C year$^{-1}$, with a mean warming trend of 0.03 °C year$^{-1}$, which is similar to findings of previous studies, e.g., Kattel and Yao [9]; Khatiwada et al. [8]; Shrestha et al. [7], focused on the Central Himalayan region. The highest rate of increase in temperature was observed at the Pokhara station (0.035 °C year$^{-1}$) and Jomsom station (0.029 °C year$^{-1}$) from 1971–2015. While, in the case of the entire period, the highest rate of increasing mean annual temperature was observed at Nuwakot station (0.015 °C year$^{-1}$).

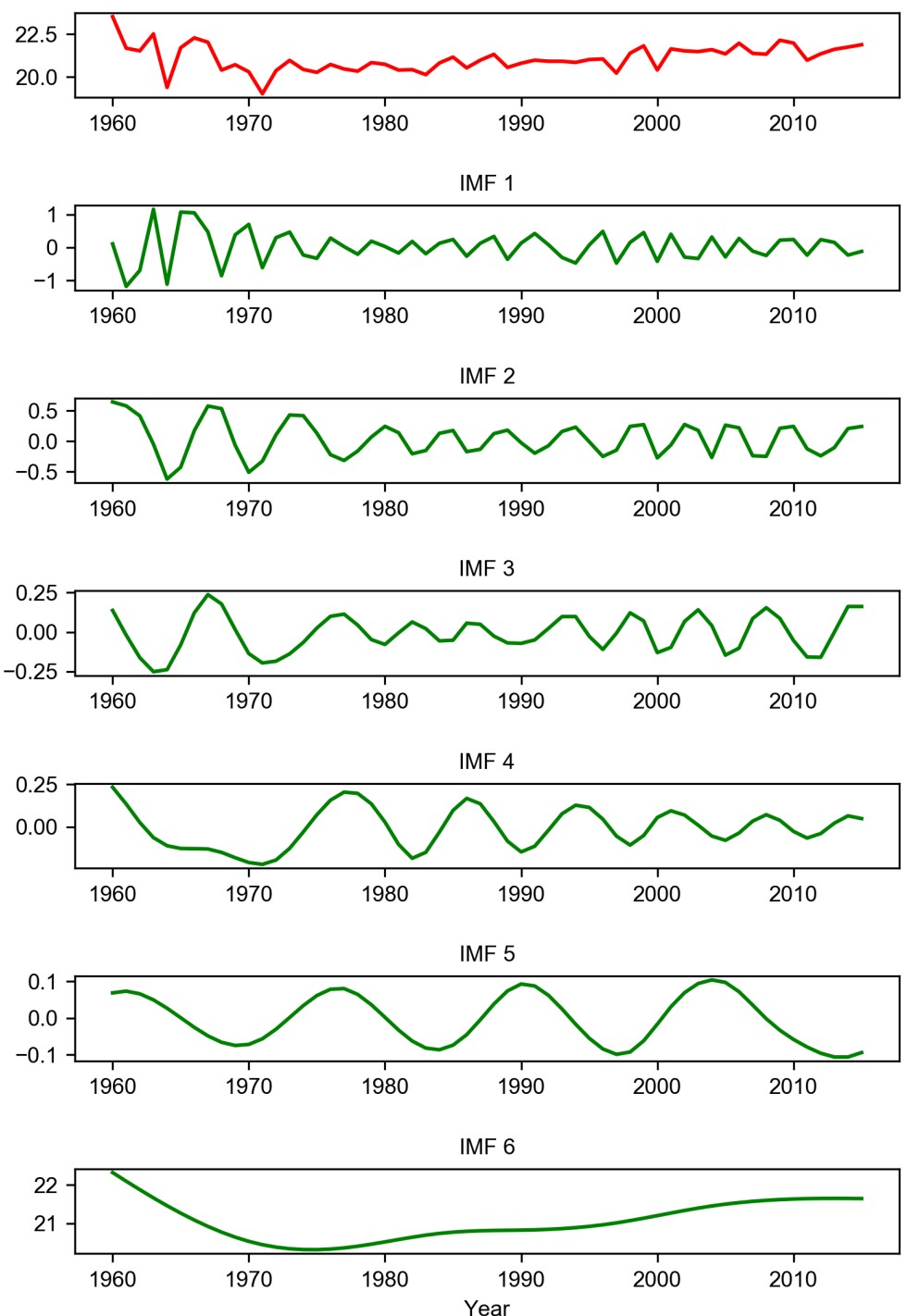

**Figure 6.** EMD decomposition of the annual mean temperature (1960–2015) in Pokhara station, with IMFs components (IMF1–IMF6) and the final residue or preliminarily identified trend.

The Mann–Kendall test was also used for the period 1970–2015 and 1980–2015 for each station and a statistically significant (95%) warming trend of 0.031 and 0.032 °C year$^{-1}$ was observed, respectively. The number of stations with statistically significant rates was increased to five, except Chapkot station for the period 1970–2015. We investigated temperature trends at different elevations for the different time periods. This analysis did not show a clear relationship for all time periods and stations analyzed. However, a comparison of the highest (Langtang) and lowest (Chapkot) elevation stations for the period 1980–2015

revealed a higher warming rate at Langtang (0.044 °C year$^{-1}$) compared to Chaptkot (0.013 °C year$^{-1}$). Additionally, the highest warming rate (0.04 °C year$^{-1}$) at Langtang station was observed from 1993 to 2015 compared to other low altitude stations; however, this rate is not statistically significant. Similar values were also reported by Salerno et al. [50]. The insignificant statistical test can be attributed to high variability, significant gap and instrumental error, involved with high altitude observations. Nevertheless, this is a sign of a high warming rate in the high mountain region and a lower warming rate in lowland areas. However, the test results for 1970–2015, 1990–2015 and 2000–2015 do not show a clear relationship between elevation and higher warming rates. Similar results are also shown by Kattel and Yao [9] and You et al. [4], and this might possibly be due to the small number of stations at different elevation zones. This needs to be verified using several stations in the region.

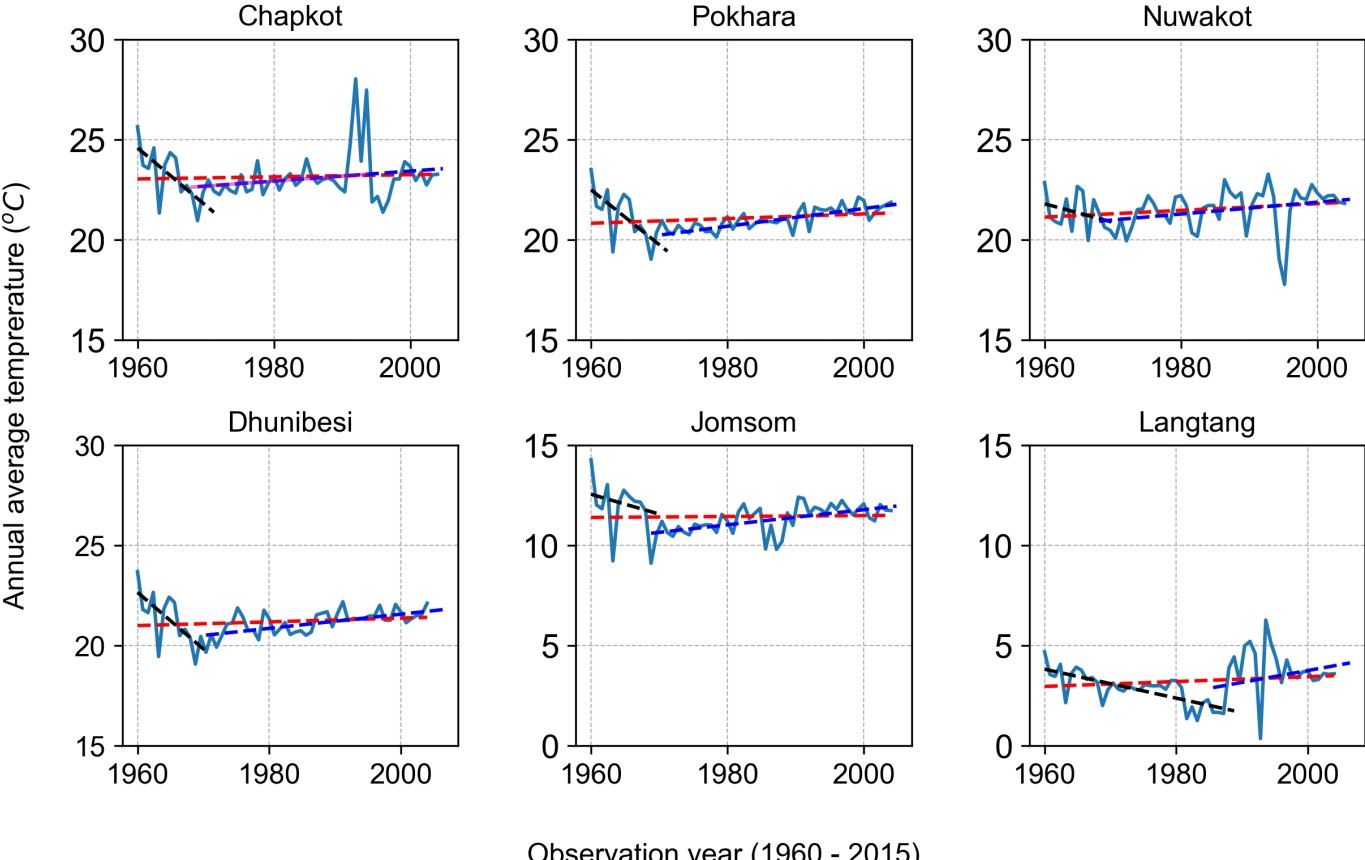

**Figure 7.** Historical mean annual temperature trend in the Narayani River basin for six stations. The trend line for the earlier period for each station is denoted by black lines and the later period is denoted by blue lines, while the overall trend is denoted by red lines.

The results obtained from the Mann-Kendall test for Pokhara and Dhunibesi stations revealed a significant cooling trend during the period ranging from 1960 to 1970–1972, which is similar to previous findings, e.g., Liu et al. [24], Shrestha et al. [7] and others. Similarly, a statistically significant cooling trend was also observed at Langtang station for the period from 1960–1992. The mean cooling trend for Pokhara, Chapkot, Dhunibesi, and Langtang station was −0.18 °C year$^{-1}$; however, the cooling trend is not very significant for Jomsom and Nuwakot station in the studied period.

**Table 3.** Mean annual temperature trend (°C year$^{-1}$) in the Narayani River basin at six different stations. ''*'' represents statistically significant at 95% significance level.

| SN | Station Name | First Period | | Second Period | | Entire Period | |
|---|---|---|---|---|---|---|---|
| | | Date | Temp. Trend | Date | Temp. Trend | Date | Temp. Trend |
| 1 | Chapkot | 1960–1972 | −0.2207 * | 1973–2015 | 0.0168 | 1960–2015 | 0.0009 |
| 2 | Pokhara Airport | 1960–1970 | −0.2110 * | 1971–2015 | 0.0354 * | 1960–2015 | 0.0102 * |
| 3 | Nuwakot | 1960–1972 | −0.2145 | 1973–2015 | 0.0280 * | 1960–2015 | 0.0152 * |
| 4 | Dhunibesi | 1960–1972 | −0.2145 * | 1973–2015 | 0.0280 * | 1960–2015 | 0.0080 * |
| 5 | Jomsom | 1960–1970 | −0.0790 | 1971–2015 | 0.0294 * | 1960–2015 | 0.0016 |
| 6 | Langtang | 1960–1992 | −0.0550 * | 1993–2015 | 0.0410 | 1960–2015 | 0.0107 |

### 4.3. Seasonal Trends

The mean monthly temperature for the study period at each station (Figure 8) is the lowest in January and the highest during June–August. The lowest mean monthly temperature was found at Langtang station (−0.88 °C), followed by Jomsom station (3.71 °C) in January. The highest mean monthly temperature was found at Chapkot station (28.31 °C) in August, followed by Dhunibesi station (26.20 °C) in June. To get a better overview of the data spread from the mean, the monthly average temperature variance (square of the standard deviation) for each station was calculated (Figure 9). Higher variance of monthly average temperature was observed in the months of April and December, compared to other months. April indicates the transition between winter and summer months. Higher monthly mean variance in April is the indicator for changing the number of days in winter and summer. Hanjra and Qureshi [51], KC and Ghimire [52] also observed more frequent warmer days and less frequent cooler nights throughout Nepal. Annual maximum temperature is in an increasing trend with hot summer days, while minimum temperature is in a decreasing trend with cool winter days [53], which is reflected by the higher variance in the months of April and December.

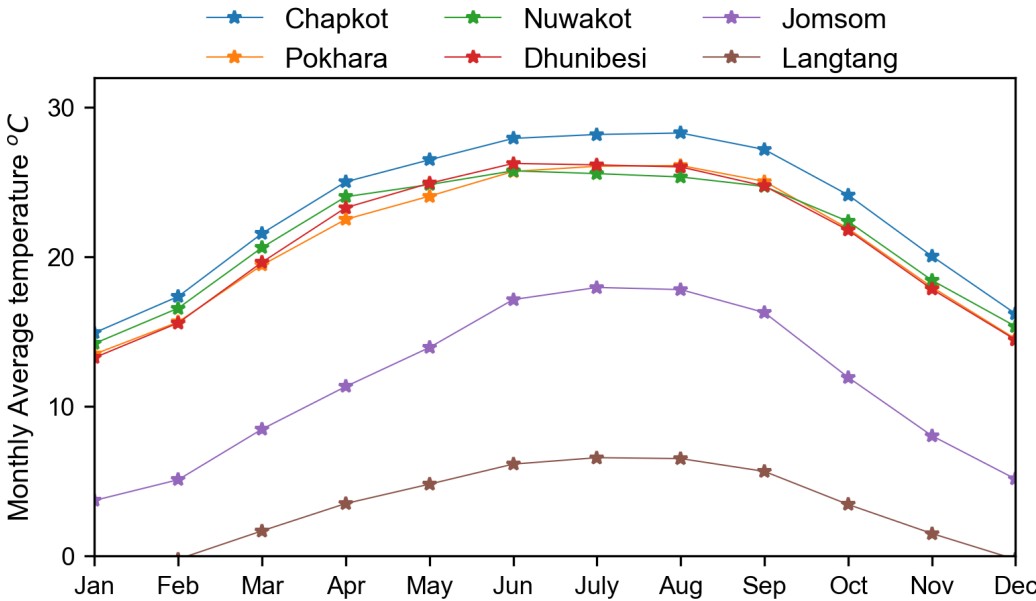

**Figure 8.** Monthly mean temperature for six stations in the Narayani River basin.

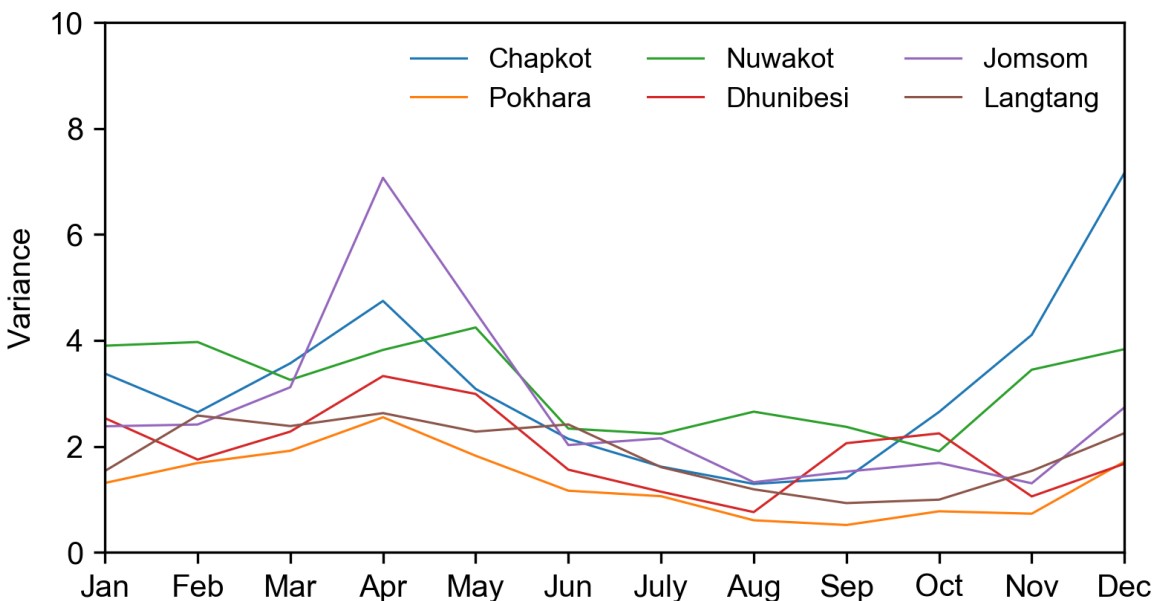

**Figure 9.** Monthly mean temperature variance at a different station.

For the seasonal study, a calendar year was divided into four different seasons [33]; winter (December–February), premonsoon (March–June), monsoon (July–August), and postmonsoon (September–November) to assess the seasonal temperature trends. The temperature at each station is symmetrical, with the highest temperature during monsoons and lowest during the winter season. The Mann–Kendall test results for seasonal trends show that the rate of increase in mean temperatures was higher during monsoon season compared to the other seasons (Figure 10)—also reported by [10] for the Tibetan Plateau between 1988–2013—and lowest in the premonsoon season—also reported by Shrestha et al. [7] in Nepal. The mean increasing rate in monsoon, winter, premonsoon and postmonsoon seasons were 0.040, 0.036, 0.034 and 0.032 °C year$^{-1}$, respectively, and those values were statistically significant at the 95% significance level. Warming rates in all seasons were consistent with observations made by Shrestha et al. [7], Khatiwada et al. [8] and Liu et al. [24] at the basin and regional scales. Significant warming trends were observed at five out of six stations in the monsoon, winter and postmonsoon seasons and at four of six stations in the premonsoon season (Table 4). The significant warming trend in winter is because of the decreasing number of winter days, which is also mention by Devkota [54] in his study. Warming trends during premonsoon, monsoon and postmonsoon seasons could be because of changes in monsoon onset time. The study presented by Sirvastava et al. [55] also indicates changes in monsoon time and its impacts on temperature trend.

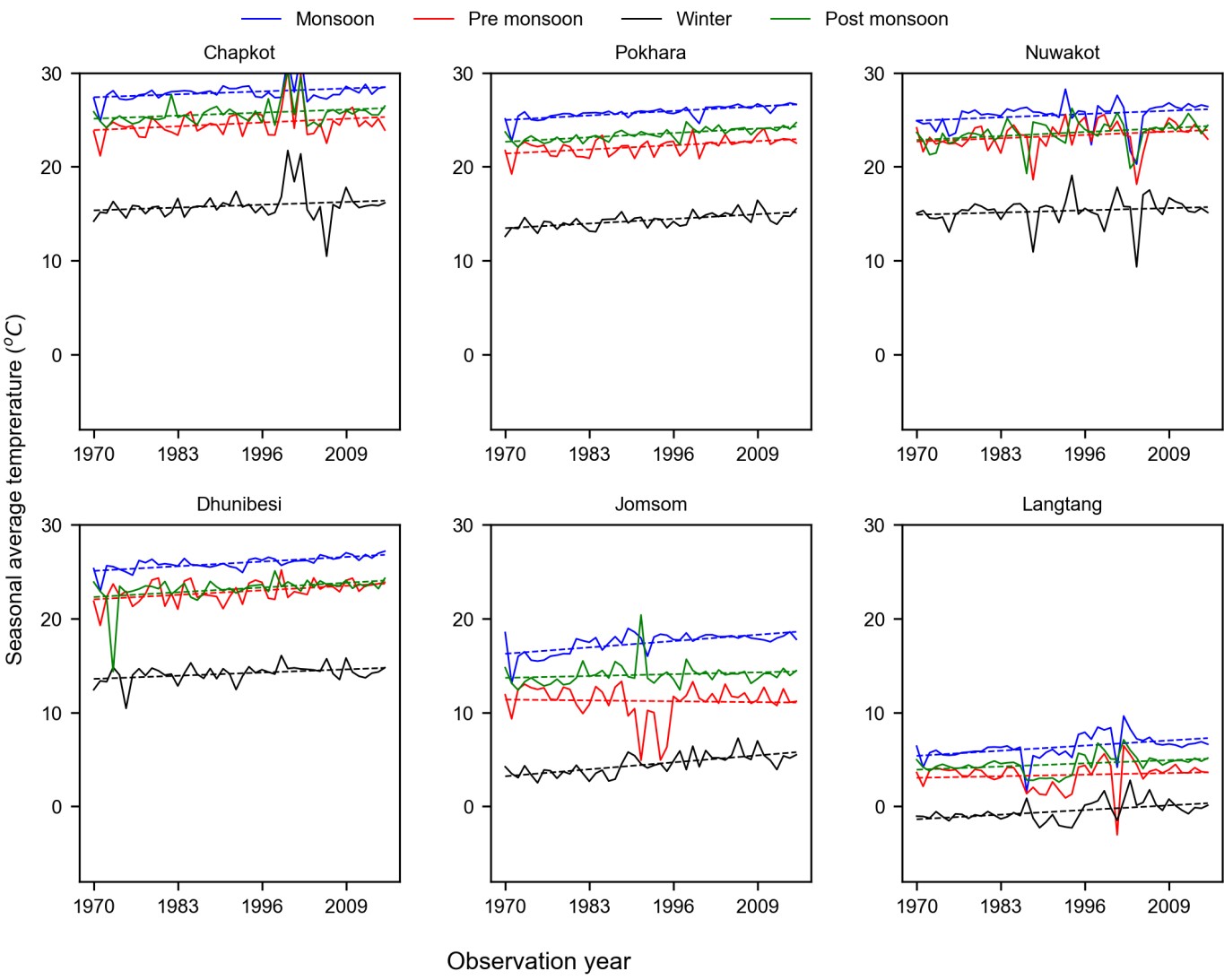

**Figure 10.** Historical mean seasonal temperature trend in the Narayani River basin for six stations.

**Table 4.** Mean seasonal temperature trend (°C year$^{-1}$) in the Narayani River basin at six different stations from 1970–2015. ''*'' represents statistically significant at 95% significance level.

| Station ID | Station Name | Date | Monsoon | Winter | Premonsoon | Postmonsoon |
|---|---|---|---|---|---|---|
| 1 | Chapkot | 1970–2015 | 0.021 | 0.019 | 0.027 | 0.021 * |
| 2 | Pokhara Airport | 1970–2015 | 0.037 * | 0.038 * | 0.034 * | 0.036 * |
| 3 | Nuwakot | 1970–2015 | 0.03 * | 0.021 * | 0.03 * | 0.036 * |
| 4 | Dhunibesi | 1970–2015 | 0.038 * | 0.024 * | 0.037 * | 0.039 * |
| 5 | Jomsom | 1970–2015 | 0.051 * | 0.057 * | −0.0080 | 0.014 |
| 6 | Langtang | 1970–2015 | 0.042 * | 0.038 * | 0.015 | 0.027 * |

*4.4. Temperature Gradients*

Temperature gradients were calculated using temperature data from different elevations within the basin, following Immerzeel et al. [56]. A gradual decrease in mean annual temperature from south to north was observed from six stations at different elevations. The mean annual temperature was highest at Chapkot station (460 m a.s.l.) and lowest at Langtang station (3800 m a.s.l.). The lapse rates were calculated from linear regression

between mean temperature and elevation, and were found to be $-0.0060\,^\circ\text{C}\,\text{m}^{-1}$, with a high value coefficient of determination (Figure 11), which is similar to values reported by Baral et al. [25], Immerzeel et al. [56] and Takahashi [57] in the Langtang Valley of the same basin. The steepest temperature gradient ($-0.0064\,^\circ\text{C}\,\text{m}^{-1}$) was observed during the premonsoon season, which is also consistent with observations from Baral et al. [25] and Immerzeel et al. [56] in Langtang Valley, a mountainous part of the Narayani River basin. The largest ($-0.0052\,^\circ\text{C}\,\text{m}^{-1}$) and least negative temperature gradient was observed during the winter season, whereas postmonsoon and monsoon seasons had the same value of lapse rates ($-0.0062\,^\circ\text{C}\,\text{m}^{-1}$). However, very high lapse rates of $-0.0013\,^\circ\text{C}\,\text{m}^{-1}$ and $-0.0009\,^\circ\text{C}\,\text{m}^{-1}$ for the monsoon and winter seasons, respectively, were reported by Chand et al. [58] between Langtang station and the Lirung debris-covered glacier, located at about 4100 m a.s.l. The discrepancies between different seasons were due to differences in relative humidity [56]) and incoming solar radiation [25]. This may be further influenced by the wind circulation direction in different seasons, which may vary according to land use type [59]. The more gradual gradients of temperature between the debris-covered glacier and lowland areas were due to the strong influence of surface heating during the daytime to near-surface air temperature [58]. This indicates a strong relationship between temperature and elevation and is useful for the prediction of temperature for high mountain areas, where there is lack of observation.

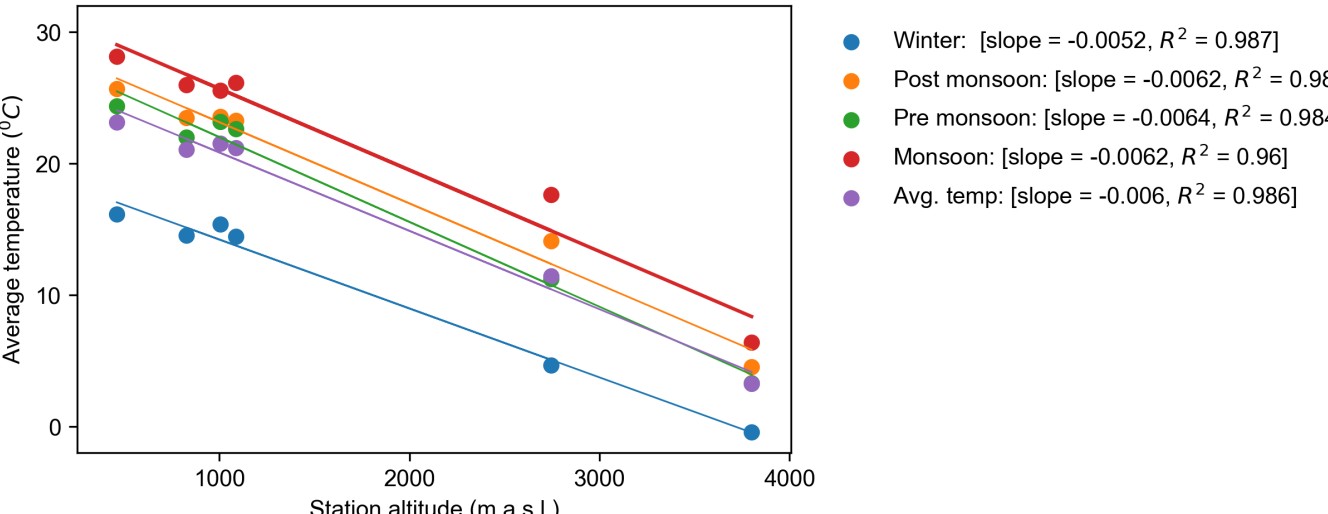

**Figure 11.** Seasonal mean temperatures at six stations plotted against elevation for the study period (1960–2015).

### 4.5. Comparison of Station Data with Freely Available Global Climate Dataset

Comparison of the mean monthly temperature from different modeled dataset (GLDAS 2.0, CHELSA 1.2) shows little difference in mean monthly temperature at Pokhara and Chapkot (Figure 12). Mean monthly temperatures were found to be lowest from GLDAS 2.0 for Jomsom and from CHELSA 1.2 for Pokhara and Nuwakot. There is no significant difference between mean monthly temperature values from WorldClim 2.0 (1970–2000) and station data (1979–2013). Data from WorldClim are closest in agreement with station data, which are quite obvious, as WorldClim 2.0 is principally based on station data.

MAAT values were estimated using monthly temperature values for all six locations using Worldclim 2.0, CHELSA, station data set (1970–2000), station data set (1979–2013), station data set (2001–2010) and GLDAS 2.0 data set (2001–2010) (Figure 13). It can be observed that the MAAT value obtained from station data set (1970–2000), station data set (1979–2013) and station data set (2001–2010) are similar for all locations. Compared to GLDAS 2.0 and Chelsa, MAAT obtained using WorldClim 2.0 is closer to the values obtained from the station data set for Dhunibesi, Jomsom, Chapkot and Langtang. Large

variations in MAAT values from different data sets is observed for Jomsom, Nuwakot, and Dhunibesi.

Monthly temperature curves obtained from the MODIS land surface temperature data set for 2014 and 2015 show that there exists a significant difference between mean monthly LST daytime and LST night-time temperatures at all stations, suggesting noteworthy diurnal variations of temperature at these locations (Figures 14 and 15). Mean monthly LST night-time temperatures were missing for the monsoon month of July in Nuwakot and August in Pokhara and Dhunibesi. Similarly, mean monthly LST night-time temperature was missing for the monsoon month of July 2014 in Chapkot and Pokhara as well. These missing values are possible due to presence of cloud cover affecting observations from MODIS satellites. Mean monthly LST night-time temperatures were often the lowest among all dataset at all stations, except Jomsom and Dhunibesi for 2014 as well as 2015. Mean monthly temperatures from LST (average) and station data were quite similar for Jomsom and Pokhara for 2014–2015. Mean monthly temperatures from LST (average) and GLDAS 2.0 were quite similar for Chapkot for 2015–2015.

Linearly fitted trend lines in scatterplots between mean monthly averages and MAAT values from WorldClim 2.0 and station data (1970–2000) show very strong correlation ($R^2 > 0.95$) between the two dataset for all stations (Figure 16).

Linearly fitted trend lines in scatterplots between mean monthly averages and MAAT values from CHELSA 1.2 and station data (1979–2013) also show very strong correlation ($R^2 > 0.94$) between the two data sets for all stations (Figure 17).

Linearly fitted trend lines in scatterplots between mean monthly averages from GLDAS 2.0 (2001–2010) and station data (2001–2010) show very strong correlation ($R^2 > 0.94$) between the two dataset for Jomsom, Dhunibesi and Pokhara, while a reasonably strong correlation is shown between the two data sets for Nuwakot ($R^2 = 0.78$) and Chapkot ($R^2 = 0.73$) (Figure 18).

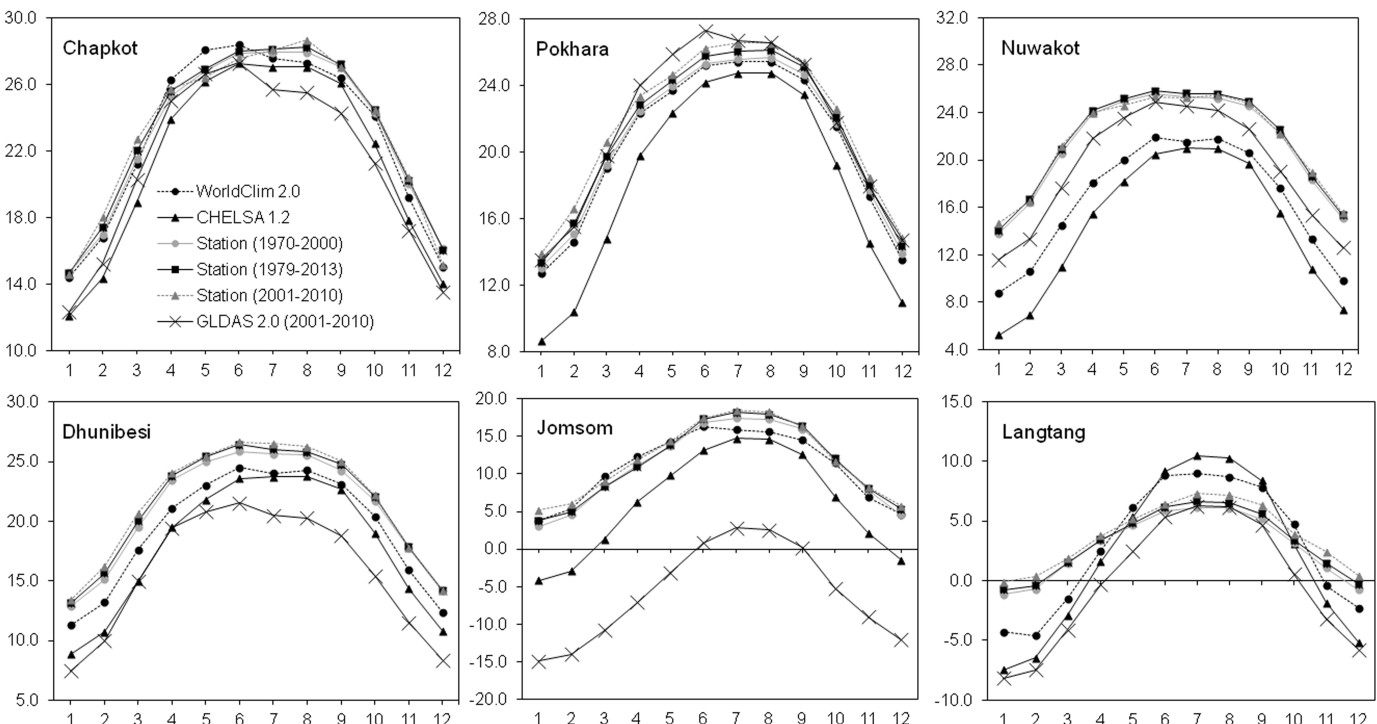

**Figure 12.** Comparison of mean monthly temperature from WorldClim 2.0, CHELSA Version 1.2, station data (1970–2000), station data (1979–2013), station data (2001–2010) and GLDAS 2.0 dataset. The numbers 1–12 on X-axis represent months in a year, where 1 means January and 12 means December. Values on Y-axis represent temperature in degrees Celsius.

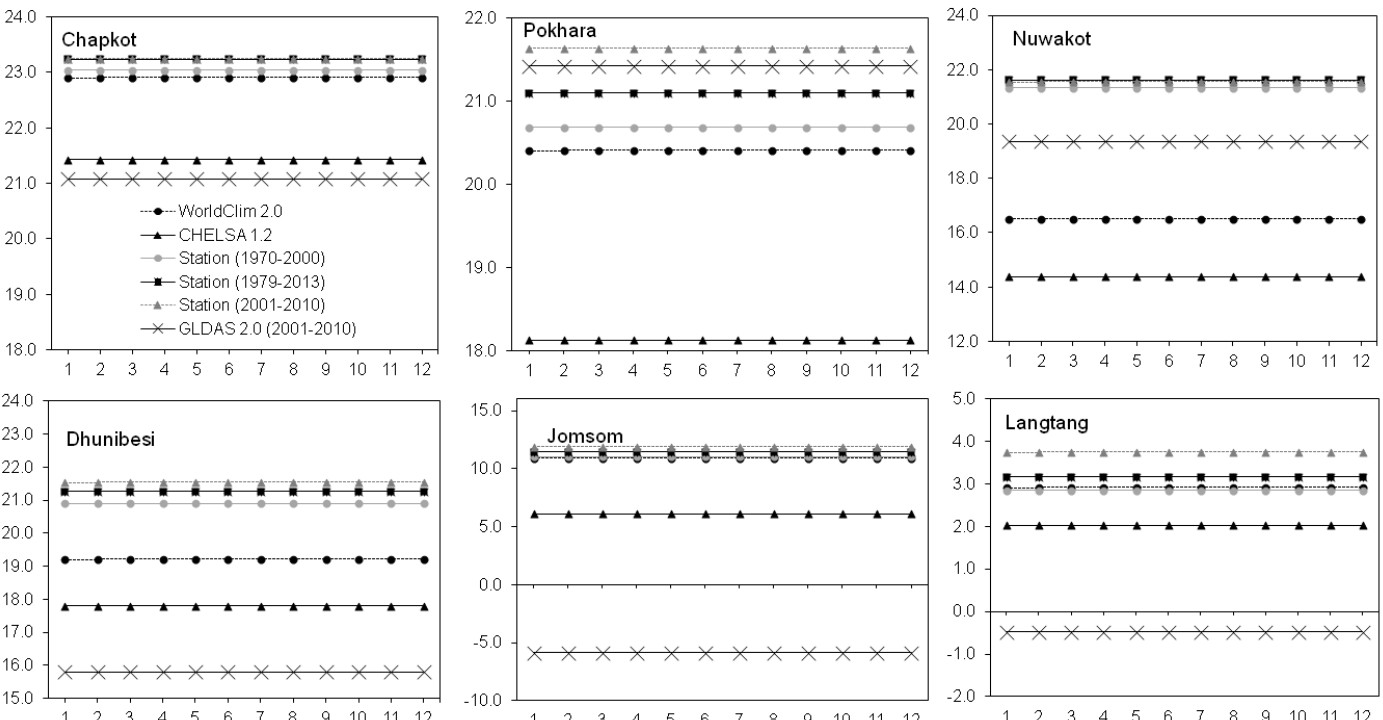

**Figure 13.** Comparison of MAAT from WorldClim 2.0, CHELSA Version 1.2, station data (1970–2000), station data (1979–2013), station data (2001–2010) and GLDAS 2.0 dataset. The numbers 1–12 on X-axis represent months in a year, where 1 means January and 12 means December. Values on Y-axis represent temperature in degrees Celsius.

Linearly fitted trend lines in scatterplots between mean monthly averages from MOD11A2 LST average of daytime and night-time values (2015), and station data (2015) show a very strong correlation ($R^2 > 0.94$) between the two dataset for Nuwakot, Chapkot and Dhunibesi, a very strong correlation ($R^2 > 0.89$) between the two dataset for Jomsom ($R^2 = 0.89$), and reasonably strong correlation between the two dataset for Pokhara ($R^2 = 0.79$) and Langtang ($R^2 = 0.76$) (Figure 19). Similarly, mean monthly averages from MOD11A2 LST of daytime and night-time values (2014) and station data (2014) also showed a very strong correlation ($R^2 = 0.90$) for Pokhara ($R^2 = 0.93$), Nuwakot ($R^2 = 0.90$), Dhunibesi ($R^2 = 0.95$), Chapkot ($R^2 = 0.88$) and Langtang ($R^2 = 0.85$), while reasonably strong correlation for Jomsom ($R^2 = 0.78$).

Overall, the gap-filled data after interpolation is in good agreement with climate data sets from different freely available sources. The consistency between gap filled data set and freely available climate data set justifies the procedure for gap filling, which is usually necessary for the mountain regions, where the malfunctioning of weather stations is usually frequent. This also shows that freely available climate data sets are very good alternatives for remote areas that are otherwise lacking a data set from well monitored weather stations.

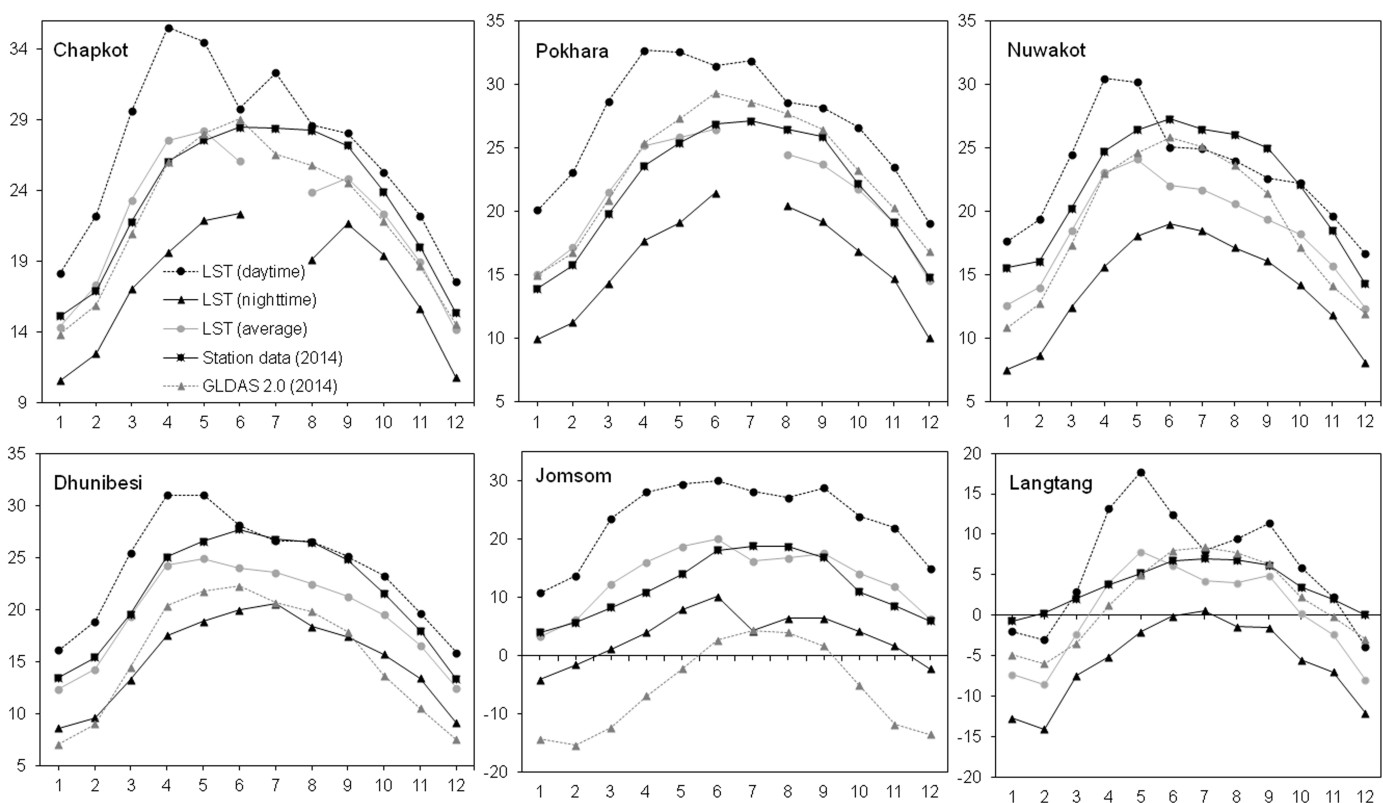

**Figure 14.** Comparison of LST (daytime), LST (night-time), LST (average), station data (2014) and GLDAS 2.0 data (2014) for all six stations. The numbers 1–12 on X-axis represent months in a year, where 1 means January and 12 means December. Values on Y-axis represent temperature in degrees Celsius.

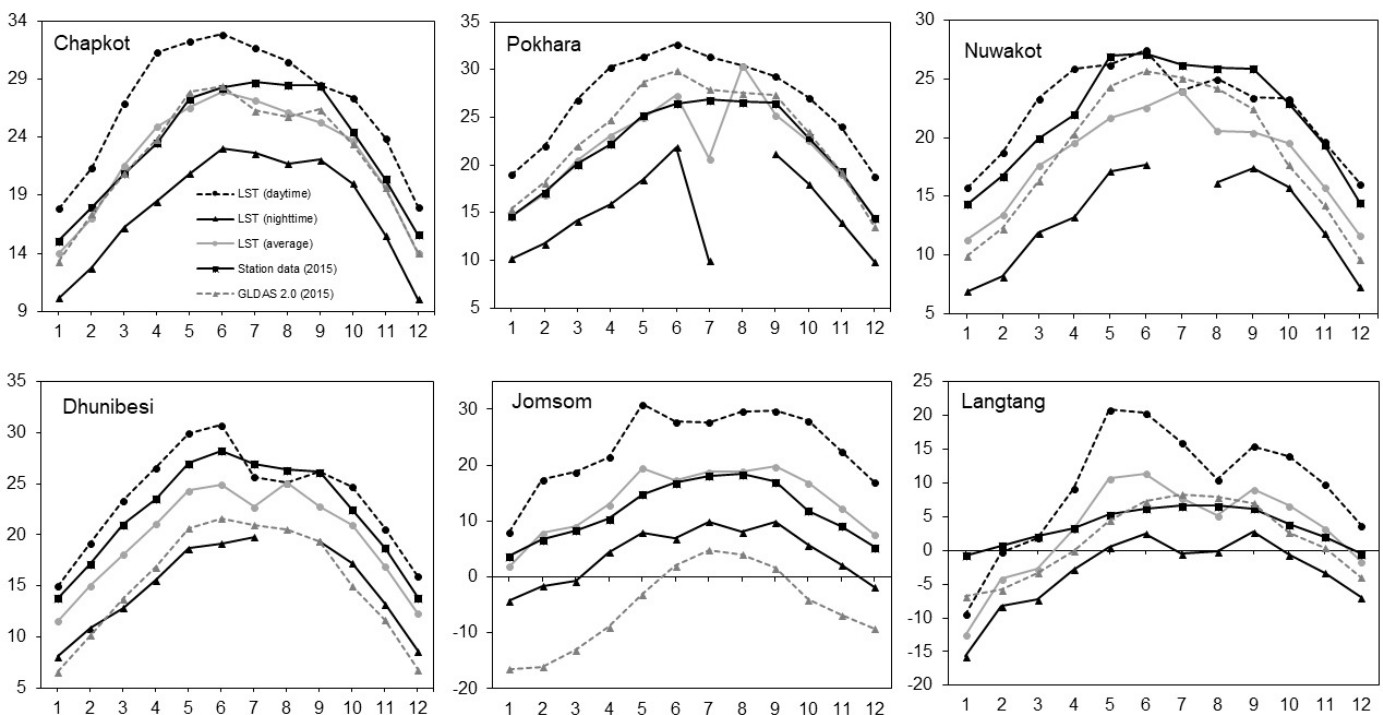

**Figure 15.** Comparison of LST (daytime), LST (night-time), LST (average), station data (2015) and GLDAS 2.0 data (2015) for all six stations. The numbers 1–12 on X-axis represent months in a year, where 1 means January and 12 means December. Values on Y-axis represent temperature in degrees Celsius.

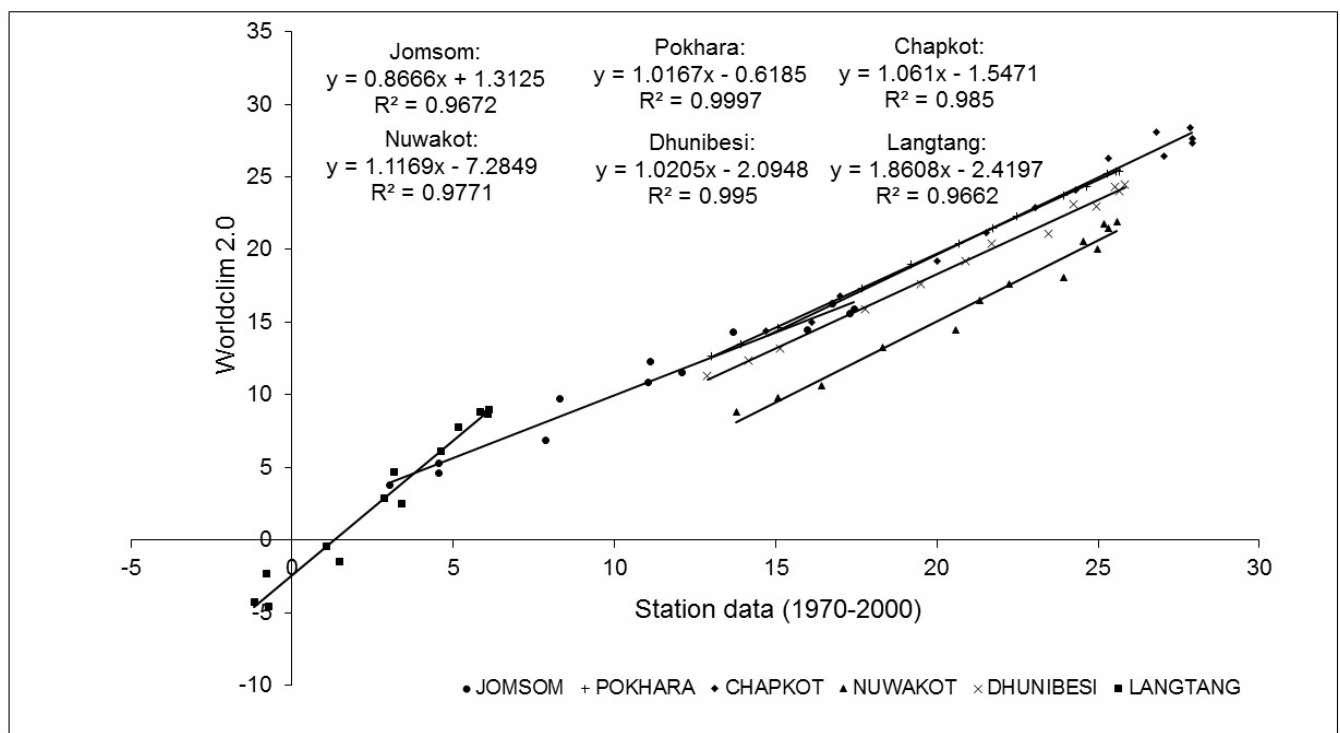

**Figure 16.** Scatterplots between WorldClim 2.0 and station dataset (1970–2000) for all six stations with fitted linear trend lines. Values on X-axis and Y-axis represent temperature in degrees Celsius. The equation of linear regression and R-squared values for all six stations are shown in the figure.

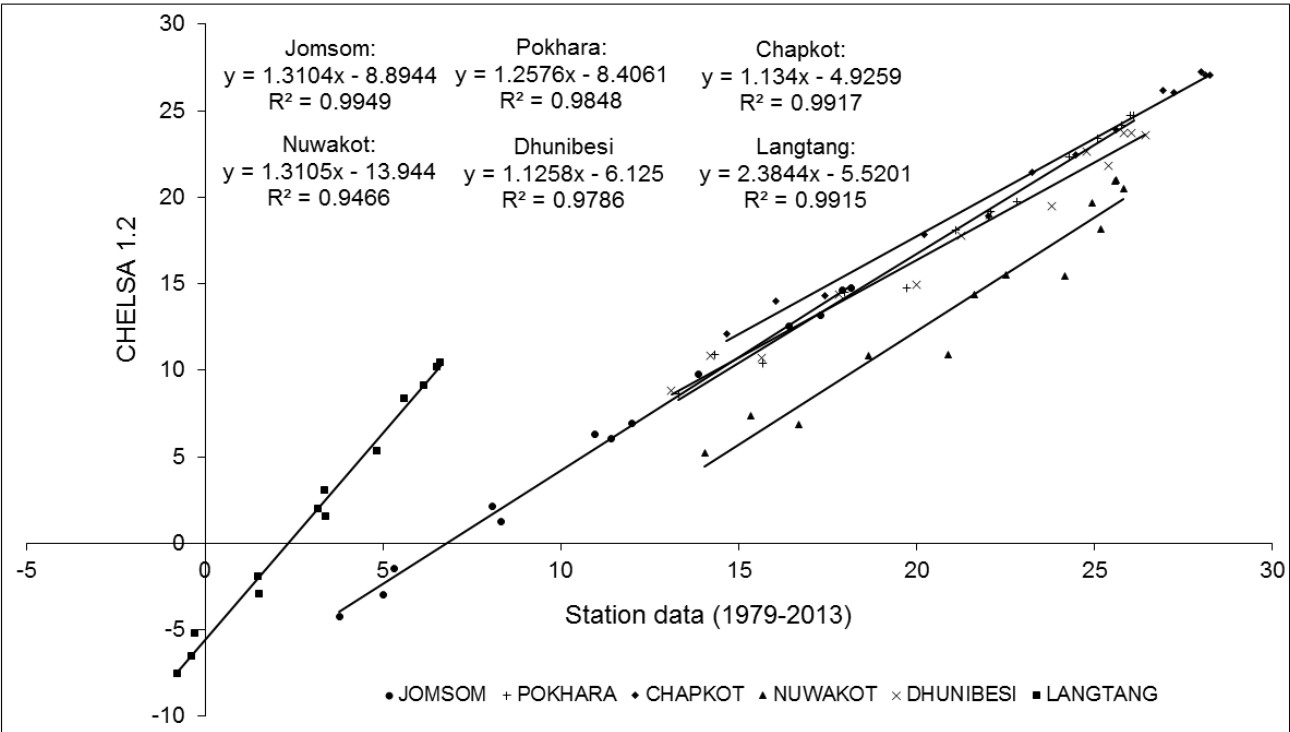

**Figure 17.** Scatterplot between CHELSA 1.2 and station data set (1979–2013) for all six stations with fitted linear trend lines. Values on X-axis and Y-axis represent temperature in degrees Celsius. The equation of linear regression and R-squared values for all six stations are shown in the figure.

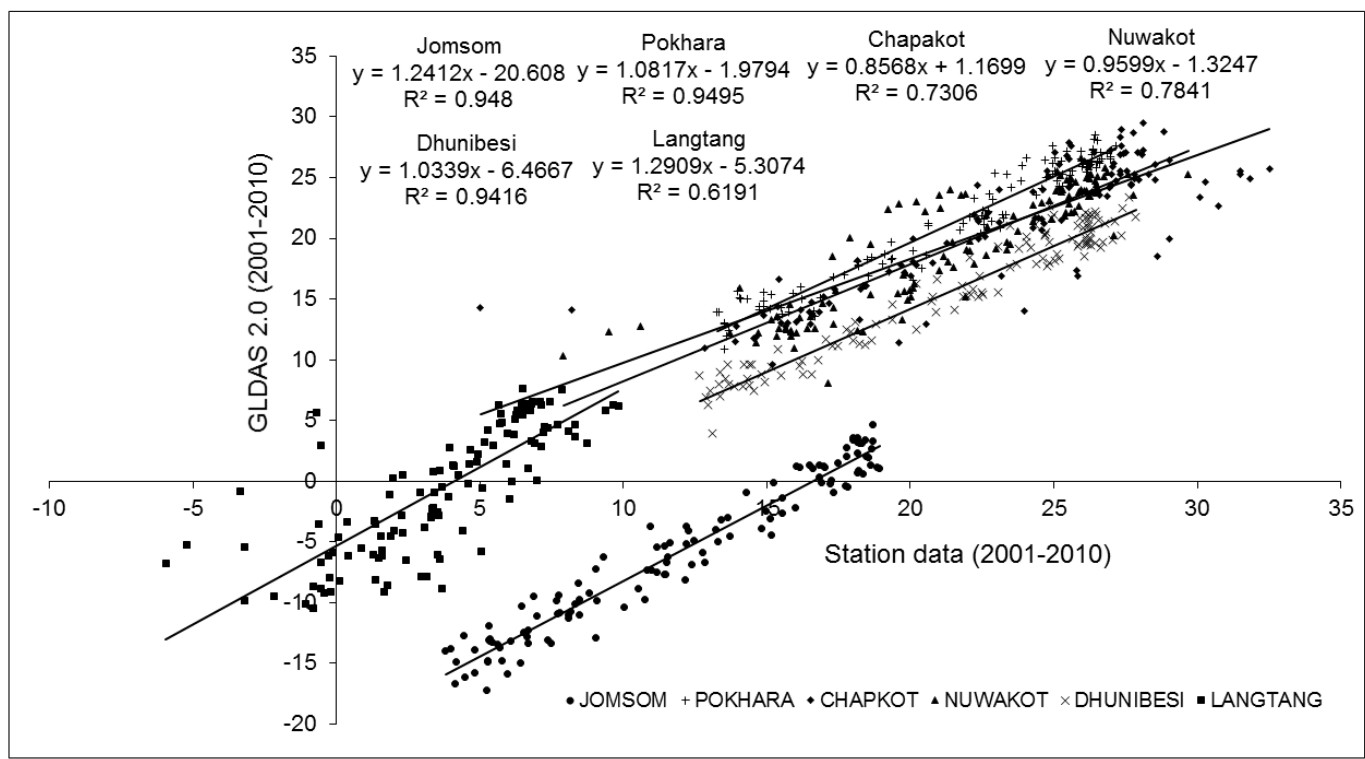

**Figure 18.** Scatterplot between GLDAS 2.0 and station data set (2001–2010) for all six stations with fitted linear trend lines. Values on X-axis and Y-axis represent temperature in degrees Celsius. The equation of linear regression and R-squared values for all six stations are shown in the figure.

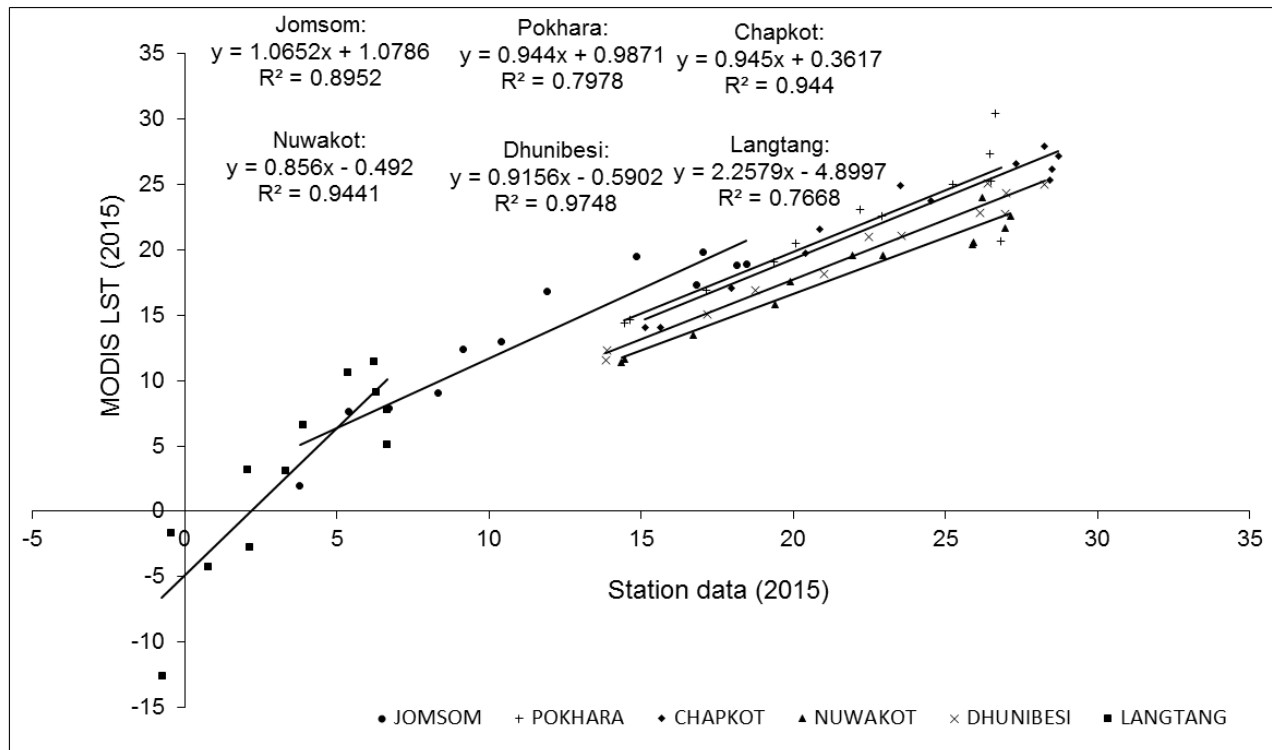

**Figure 19.** Scatterplots between MOD11A2 LST (average) and station data set (2015) for all six stations with fitted linear trend lines. Values on X-axis and Y-axis represent temperature in degrees Celsius. The equation of linear regression and R-squared values for all six stations are shown in the figure.

## 5. Conclusions

This study contributes to the understanding of annual and seasonal temperature trends for the last fifty years (1960–2015) and several intermediate periods in the Narayani River basin of Nepal. It also demonstrates that the application of simple and multiple linear regression methods with ordinary least squares assumption is efficient to fill gaps in temperature dataset. The gap-filled daily mean temperature data set is prepared for the duration of the period 1960–2015. Further, this study applied EMD to identify the trend break and Mann–Kendall test to assess the mean temperature trend. Strong to very strong correlation between the gap-filled data after interpolation with temperature data sets from different freely available sources enhances the reliability of the gap-filling procedure. Simultaneously, it also suggests that freely available temperature data sources can be equally useful for studies in mountainous river basins where station-based data sources are scarce.

Based on analysis of long-term temperature data sets from six different stations located within the Narayani River basin, this study observed the following:

1. The mean annual temperature trend shows a trend break in the 1970s for most of the stations. After the 1970s, the mean temperature increased at a statistically significant rate in the majority of stations.
2. The rate of increase in mean annual temperature ranges from 0.028 to 0.035 $^\circ$C year$^{-1}$ with a mean warming trend of 0.03 $^\circ$C year$^{-1}$.
3. The highest increase in annual mean temperature is recorded in the monsoon season, followed by the winter season, postmonsoon season and premonsoon season, respectively.
4. The temperature lapse rate with altitude is 0.006 $^\circ$C m$^{-1}$ in the Narayani River basin with the steepest value in the premonsoon season.
5. The lapse rates calculated here are useful for temperature prediction in the higher Himalayan region where observations are scarce.

In general, an increasing trend in temperature, which is higher than the global average, makes glaciers and snow-covered areas in this region vulnerable to widespread melt, which has a significant impact on the hydrological cycle. In this study, six stations were assumed as representative stations for the whole basin as they are located at different elevations and physiographic zones; however, the inclusion of other available observed, reanalysis, and satellite dataset for this basin can be very useful to represent each micro-climatic zone within this basin.

**Author Contributions:** M.B.C. and B.C.B. designed the analysis. M.B.C. and B.C.B. performed the experiments, and analyzed the data. M.B.C. and B.C.B. wrote the manuscript with input from P.B. and N.S.P. All authors have read and agreed to the published version of the manuscript.

**Funding:** This research received no external funding.

**Acknowledgments:** We are thankful to the Department of Hydrology and Meteorology, Government of Nepal (DHM, GoN) for providing massive data.

**Conflicts of Interest:** The authors declare that they have no conflict of interest.

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
