# Peer review of "Trend Analysis of Temperature Data for the Narayani River Basin, Nepal"

_sci, doi:10.3390/sci3010001_

Round 1
Reviewer 1 Report
Paper provides useful information of temperature variation in Himalayan region. Paper showed temperature trend in five recent decades. Temperature significantly increased, and break point is around 1970s. This is new discovery, it hopefully contribute to enhance understanding of climate of Himalayan region. I believe that it should be published in the Journal after improvement.
There are some minor comments:
In abstract, the phrase “spatiotemporal dynamics” should be change to “spatiotemporal variation”
In Figure 3, at some stations, why temperature dramatically changes in around 2000. It is quite different with other period. You can give some evidence or reference of controlling reason to confirm that your data is correct.
The data in Table 2. Please discuss benefits of the data in this table for your research goals. In other words, this data is for what purpose.
In Section 3.3; “Each station was treated separately in order to fill the missing values. Independent variables for the model were selected based on the highest correlation with surrounding station in the model development process to fill the gap.” I think two sentences are conflict. Since, to fill the missing values, you need data of surrounding stations. But the first said, you did that separately.
Station ID 1000, original data was only a few, while data re-constructed too long. Whether the data quality is guaranteed or not. Thus, it can influence the following analysis.
I think Section 3.6 should be placed next to Section 3.3. I think it not needed to organize in separated Section. You can briefly describe.
In figure 9, show unit of variable.
In Section 4.4 (temperature gradient), please explain more specifically how you calculate that.

Reviewer 2 Report
Chand et al. investigate the spatio-temporal temperature dynamics in Narayani river basin, located in Nepal. They analyze temperature data from six stations, distributed at different elevations within the basin, data availability ranging from 1960-2015. Since all the stations present significant data gaps, they use multiple regression and empirical mode decomposition to fill these gaps. The comparison between filled data and temperature datasets provided by different freely available sources shows a good correlation, supporting the reliability of the filling procedure. The data analysis shows a significant increasing temperature trend after 1970s in the majority of the stations (mean warming trend of 0,03°C per year), with higher rate of increase during monsoon season and winter. Furthermore, Chand et al. finds a relation between temperature and elevation, showing a temperature lapse rate with altitude of 0,006°C per meter within the investigated basin, which can be very useful considered the lack temperature data in the higher Himalayan region.
Overall I found the manuscript well written and it highlights once again a significant issue of our age, climate change and global warming, focusing on an area very sensitive to the global temperature increase like the Himalayan region. The analyses and interpretations are sound, but in my opinion the discussion about some points should be expanded. In my view, the manuscript is suitable for publication in Sci with a minor to moderate revision.
Here it follows a list of suggestions and observations the authors might want to consider:
- Data clearly show that, until 1970, there is a significant cooling trend and that this cooling trend last longer in time at many of the measuring stations. Can you explain this behavior? Can you provide any hypothesis regarding the cooling trend preceding the clear warming trend that follows?
- Figure 1: I would appreciate if you could highlight the Narayani River in the map.
- Paragraph 3.1: “data gaps more than 5 years” I think it would be better to write “data gaps longer than 5 years”.
- Paragraph 3.2: even if it is quite clear, I think you should briefly explain why the skewness of the data follows from the statistical analysis.
- Figure 2/3/5/6/8/9: please adjust the dimension of the figures in order to make them more homogeneous with the others figures and the rest of the manuscript.
- Figure 4: I think that all the boxes in figures 4 should be of the same dimension in order to make them comparable, also the last one.
- Paragraph 3.5: “Shrestha et al. [18] and [35].” It seems that a citation is missing.
“that is, the MK test is a non-parametric (distribution-free) test.” I think there is an error.
- Paragraph 4.1: both in Paragraph 4.1 and Figure 6 you show and describe all the components obtained with the EMD decomposition, but then you just discuss and focus on the last, and probably more interesting, one (IMF 6). I think that, since you show all the components of the decomposition, you should spend some words on each one of them, explaining the meaning of the results you obtained, at least briefly.
- Paragraph 4.2: as you highlight, most of the measuring stations show a trend break in between 1970 and 1979, while the measuring station of Langtang shows the same trend break in 1993. Can you explain that? Or does this just depend on the lack of data for the measuring station of Langtang before 1988? In both cases, I think you should discuss this point.
- Paragraph 4.3: “Higher monthly mean variance in April is the indicator for changing the number of days in winter and summer. Hanjra and Qureshi [47], KC and Ghimire [48] also observed more frequent warmer days and less frequent cooler nights throughout Nepal. Annual maximum temperature is in an increasing trend with hot summer days while minimum temperature is in decreasing trend with cool winter days [49] which is reflected by the higher variance in the month of April and December.” I think this is a very interesting point and the discussion should be expanded.
“Significant warming trends were observed at five out of six stations in the monsoon, winter and post-monsoon seasons and at four of six stations in pre-monsoon season (Table 4).” Can you suggest an explanation about that?
- Paragraph 4.4: “The discrepancies between different seasons are due to differences in relative humidity [51]) and incoming solar radiation [24].” I think it would be useful to explain how relative humidity and solar radiation affect temperature causing the highlighted difference between different seasons.
- Figure 12: I think that MAAT should not be displayed as the last point of the plot, but it should be shown by itself. This representation might be misleading. Please, add the axes labels both here and in Figure 13.
- Paragraph 4.5: I think that MAAT values should not be considered when fitting the trend lines shown in Figures 14-15-16-17. It won’t change that much but it seems uncorrect.

Author Response
General comments: Chand et al. investigate the spatio-temporal temperature dynamics in Narayani river basin, located in Nepal. They analyze temperature data from six stations, distributed at different elevations within the basin, data availability ranging from 1960-2015. Since all the stations present significant data gaps, they use multiple regression and empirical mode decomposition to fill these gaps. The comparison between filled data and temperature datasets provided by different freely available sources shows a good correlation, supporting the reliability of the filling procedure. The data analysis shows a significant increasing temperature trend after 1970s in the majority of the stations (mean warming trend of 0,03°C per year), with higher rate of increase during monsoon season and winter. Furthermore, Chand et al. finds a relation between temperature and elevation, showing a temperature lapse rate with altitude of 0,006°C per meter within the investigated basin, which can be very useful considered the lack temperature data in the higher Himalayan region. Overall I found the manuscript well written and it highlights once again a significant issue of our age, climate change and global warming, focusing on an area very sensitive to the global temperature increase like the Himalayan region. The analyses and interpretations are sound, but in my opinion the discussion about some points should be expanded. In my view, the manuscript is suitable for publication in Sci with a minor to moderate revision. Dear Reviewer, Thank you very much for providing valuable suggestions for our manuscript. We are very much thankful to the reviewer who gave insightful comments and suggestions, which helped us to improve the manuscript. Your suggestions really helped to increase the quality of the manuscript. We tried our best to address all suggestions and comments suggested by you. We are also thankful to you for your appreciation to our work and being positive for the publication. The responses to each comment are provided below. Here it follows a list of suggestions and observations the authors might want to consider: - Data clearly show that, until 1970, there is a significant cooling trend and that this cooling trend last longer in time at many of the measuring stations. Can you explain this behavior? Can you provide any hypothesis regarding the cooling trend preceding the clear warming trend that follows? Most of the stations showed the cooling trend before the 1970s, however, station i.e Langtang shows a cooling trend till 1993. The original dataset of Langtang was ranging from 1988-2008 and we filled the data before and after this period using the simple and multiple linear regression methods. The earlier period was dominated by the cooling trend and it might have influenced the data filling method where there was a continuous gap before 1988 (1993). And, a minor variation of trend break year might be influenced by the variation local weather condition. Usually, we understood that carbon emission before the 1970s was limited and it was increased rapidly after this which led to rapid global warming. This effect is reflected in our results. Figure 1: I would appreciate if you could highlight the Narayani River in the map. We have incorporated the river network of the Narayani River basin in the modified version of the manuscript. Paragraph 3.1: “data gaps more than 5 years” I think it would be better to write “data gaps longer than 5 years”. Paragraph 3.2: even if it is quite clear, I think you should briefly explain why the skewness of the data follows from the statistical analysis. Figure 2/3/5/6/8/9: please adjust the dimension of the figures in order to make them more homogeneous with the other figures and the rest of the manuscript. Figure 4: I think that all the boxes in figures 4 should be of the same dimension in order to make them comparable, also the last one. Thank you very much for your comments, we have revised the manuscript per your suggestions. Similarly, we have added the sentence "Similar observed skewness from all station indicates similar seasonality in all stations" in the revised manuscript We have adjusted the figure size, their fonts, and dimensions for all figure in our revised manuscript. Paragraph 3.5: “Shrestha et al. [18] and [35].” It seems that a citation is missing. “that is, the MK test is a non-parametric (distribution-free) test.” I think there is an error. Paragraph 4.1: both in Paragraph 4.1 and Figure 6 you show and describe all the components obtained with the EMD decomposition, but then you just discuss and focus on the last, and probably more interesting, one (IMF 6). I think that, since you show all the components of the decomposition, you should spend some words on each one of them, explaining the meaning of the results you obtained, at least briefly. We feel sorry for not making a clear citation. Now Xu et al. (2017) is added to make it clear. IMF6 was the final output (IMF), from which we can find out the trend break (either monotonically downward or monotonically upward). The other IMFs were intermediate output and from which we cannot identify the just two decompositions. However, those results are useful when we were interested in a short-term anomaly of climate in response to different climatic events. Paragraph 4.2: as you highlight, most of the measuring stations show a trend break in between 1970 and 1979, while the measuring station of Langtang shows the same trend break in 1993. Can you explain that? Or does this just depend on the lack of data for the measuring station of Langtang before 1988? In both cases, I think you should discuss this point. Thank you for your critical reviewing the manuscript. You are right and your suggestion is appreciated. Most of the stations showed the cooling trend before the 1970s, however, station i.e Langtang shows a cooling trend till 1993. The original dataset of Langtang was ranging from 1988-2008 and we filled the data before and after this period using the simple and multiple linear regression methods. The earlier period was dominated by the cooling trend and it might have influenced the data filling method where there was a continuous gap before 1988 (1993). Paragraph 4.3: “Higher monthly mean variance in April is the indicator for changing the number of days in winter and summer. Hanjra and Qureshi [47], KC and Ghimire [48] also observed more frequent warmer days and less frequent cooler nights throughout Nepal. Annual maximum temperature is in an increasing trend with hot summer days while minimum temperature is in decreasing trend with cool winter days [49] which is reflected by the higher variance in the month of April and December.” I think this is a very interesting point and the discussion should be expanded. “Significant warming trends were observed at five out of six stations in the monsoon, winter and post-monsoon seasons and at four of six stations in pre-monsoon season (Table 4).” Can you suggest an explanation about that? Thank you for your comments and suggestion. Actually, a significant warming trend in the winter is because of decreasing the number of winter days which is also indicated by Devkota (2014) in their study. And a change in temperature during monsoon, pre-monsoon and postmonsoon are could be because of changes in monsoon periods in the regions. Reference to justify this statement is now added to the revised manuscript. Now the explanation with references is added into the main text. Paragraph 4.4: “The discrepancies between different seasons are due to differences in relative humidity [51]) and incoming solar radiation [24].” I think it would be useful to explain how relative humidity and solar radiation affect temperature causing the highlighted difference between different seasons. We have tried to explain it in terms of wind circulation in a different direction during different season and weather patterns. Figure 12: I think that MAAT should not be displayed as the last point of the plot, but it should be shown by itself. This representation might be misleading. Please, add the axes labels both here and in Figure 13. We are very grateful to the reviewer for these insightful suggestions. The intention behind displaying MAAT as the last point of the plot was to provide with a graphical representation for comparison of MAAT with mean monthly temperature values for 12 months. The intention behind not showing as a separate point was to display the sharp rise of the MAAT temperature values after December. This was done to emphasize the temperature difference between the maximum and minimum monthly temperature values and also to show that the MAAT values generally lie somewhere in between the maximum and minimum monthly temperature values, generally in the range of temperatures values observed for pre-monsoon (March-April) or post-monsoon (October) months. We totally agree with the reviewer that the representation might be misleading. Therefore, the labels in X-axis have been clearly explained in the figure captions. Both axes labels have been explained in the figure captions. We tried adding axes labels in the figure itself but it seemed redundant and unnecessary. We hope that the reviewer will kindly consider this situation. Paragraph 4.5: I think that MAAT values should not be considered when fitting the trend lines shown in Figures 14-15-16-17. It won’t change that much but it seems uncorrect. We are again very grateful to the reviewer for this insightful suggestion. The basic intention behind considering MAAT values when fitting the trend lines shown in Figures 14-15-16-17 was to display that MAAT values lie in close agreement with other temperature values in the scatter plot. This would be strong evidence in support of our analysis. The next reason behind considering MAAT values when fitting the trend lines was to increase the number of points (although only by one point) for comparisons made for temperature data from MODIS, CHELSA and WorldClim 2.0 as these temperature data sets have only 12 data points in the scatter plot for comparison. However, as the reviewer has already mentioned, adding these points would not change the results of the analysis but that is one another evidence in support of our analysis that MAAT values are in close agreement with another set of data points in the scatterplot. We hope that the reviewer will kindly consider this situation.Reviewer 3 Report
General:
Overall this manuscript is well prepared. The authors analyses air temperature data for six stations in Nepal. A gap-filling process was used to interpolate missing data. The gap-filled in situ data was compared with derived products (modeled data, remotely-sensed data).
Improved language clarity would greatly enhance the manuscript.
The authors do a good job of including relevant studies of similar data over a similar. Better distinction between this study and previous studies would help the reader understand the specific contribution of this piece of work.
General comments
Introduction – very good review of existing work but I found I got lost in the details. Instead of listing each study separately is it possible to synthesize the results more generally? Or perhaps include a paragraph break to more clearly demonstrate summary of in-situ vs model-based studies?
Methods
3.3 Gap filling
Please explain or elaborate on gap filling process and use of stations. In Section 3.1 you describe how stations were selected that represent different physiographic zones and climate ranges. Given this statement how would these differences affect the gap-filling process? How are these differences addressed in the gap-filling process, if at all? What would be the impact on the ‘filled’ data?
3.4-Trend break detection
Longest record used for the EMD analysis. This makes sense but you need to state if and how the results for that station might be applicable to the other stations in your region. You specified that your stations cover different physiographic and elevation zones. Are the results obtained in the EMD analysis more a reflection of the broader climate and less so of the micro/local climate? If this is the case then you could assume what is happening at Nuwakot is probably also happening at the other stations. This is just something to consider.
Other
Was any attention paid to autocorrelation when conducting your trend analysis? If so how? If not, perhaps something to consider.
Data
MODIS – include additional references, see:
Wan Z 2013Collection 6. MODIS Land Surface Temperature Products. Users’ Guide. [and references therein]
Given the climate of the region, the absence of data when clouds are present is might be an important factor affecting the usefulness of optical satellite data. MODIS data used here are averages of all available cloud-free scenes within an 8-day period. MOD11A2 values are produced when there is a minimum of one cloud-free scene within a given 8 day period. This means that variability within that 8 day period can introduce variability in the MODIS record that is not physically real. In areas affected by cloud cover this could be a significant issue. The number of days used for each retrieval is provided in the QA layers.
3.6 – please include location and date of data access of data products either here or in the references.
Add reference to MODIS data product (example is from LP DAAC):
Wan, Z., S. Hook, G. Hulley. MOD11A2 MODIS/Terra Land Surface Temperature/Emissivity 8-Day L3 Global 1km SIN Grid V006. 2015, distributed by NASA EOSDIS Land Processes DAAC, https://doi.org/10.5067/MODIS/MOD11A2.006
4.4 - Temperature gradients
The analysis of ‘lapse rate’ isn’t too convincing as the stations are in different physiographic regions. What is really being calculated is the line between temperature and elevation from your sample of stations. Lapse rate is usually at a given location. This is probably more a matter of terminology? However, in looking at the plot (figure 11) four of the stations are around 1000 m a.s.l. so it is clustered to the left. What I find more interesting than the lapse rate is that the seasonal difference in temperature is quite small at the high elevation station compared to the other 5 (I can’t tell if the mid-elevation station has a larger spread than the low elevation stations). There is something interesting here but I don’t know if it comes across the way it is described.
4.5 – Comparison of station data with freely available global climate datasets
Data availability did not permit comparison with in-situ data and derived data over a common time period. Please state this and tell the reader more clearly how you compared the datasets. i.e. for each derived data product, in situ data for the corresponding period of record was compared. I am curious as to why only 2015 was used for MOD11A2. Data for this product are available beginning in 2000.
Given that you are comparing different time periods, and your trend analysis showed changes in temperature, would you expect to have similar average temperature values? Please provide some mention of this caveat to ensure your comparisons are properly interpreted by the reader.
Specific comments
Line numbers refer to lines of text on each page. Figures, Tables and captions not included in numbering.
Overall, suggest yr-1 instead of per year, and decade-1 instead of per decade
P1 – Abstract
L2: downstream of what?
L3: glacier ice
L5: Near surface (2m) temperature data? Specify somewhere near the beginning that you are looking at air temperature, except of course where you are looking at skin temperature.
L6: m a.s.l.
L8: remove ‘the’ from ‘the missing data’
L9: add ‘a’ to ‘a Mann-Kendall’
L9: revise ‘for testing the’ to ‘to test the’
L10: ‘trend’ should be ‘trends’ plural
L12: by ‘mean increasing trend’ do you mean the mean of all six stations? Please clarify
P1 introduction
L3: remove ‘the’ from ‘the global’
L4: ‘are significantly rising since last decade’. Suggest revising to: ‘have been rising significantly over the last decade’
L6: change by 0.12°C per decade to ‘to 0.12 decade-1’
L7: change ‘the rate of increase of temperature are different’ to ‘the rate of temperature increase is’
L8-9: note – mountain glacier melt contributes to the eustatic component of SLR. [as opposed to thermal expansion]
P2
L1: revise ‘This warming are’ to ‘this warming is’
L6-7: you mention ‘several studies’ but only once citation. Does this citation discuss multiple studies or is more than one reference necessary. COLLEEN TO CHECK
L17: revise ‘by 0.03°C’ to ‘of 0.03°C
L18: and ‘by 0.38°c’ to ‘of 0.38°C
L19: suggest revising to something along the lines of: ‘found that increases in maximum temperature (0.05 °C yr-1) were larger than those of the minimum temperature… with the largest change observed during the pre-monsoon season…’
L22: revise ‘in average temperature at a rate of’ to ‘in average temperature of’
L24: revise ‘The model’ to ‘Model’
L24: ‘the Koshi River basin’ (add ‘the’)
L25: suggest revising ‘due to rise in temperature’ to ‘due to increasing temperatures’
L27: ‘the Koshi trans-boundary…’
L45 (study area): revise ‘between latitude….85.83°E’ to ‘between 25.49°N-29.28°N and 85.02°E-85.83°E’
L47: ‘at the southern edge’
L48: km² instead of sq. km
L48-49: ‘It encompasses a total area of…, and area within Nepal is about…’ to ‘It encompasses a total area of 46,300 km², 32,104 km² of which is in Nepal’
P3
L3: ‘The upper part of this basin is covered with snow’ (add ‘is’)
L7-8: suggest revising to ‘large variations in climate over the basin’
L10: suggest revising to ‘in the monsoon season’
L15 (methods, data availability): specify/clarify if it is daily air temperature.
L16 (methods, data availability): ‘were selected (Table 1) from different …’ to ‘were selected (Table 1), each representing different…’
L19: ‘and have data gaps longer than 5 years’. Unclear if you mean none of the stations have data gaps >5yrs or if they all have data gaps longer than 5yrs.
L19: suggest revising to: ‘Langtang station has the shortest temperature record’
L21: are not uniform in what sense? Slightly ambiguous.
P4
L2: suggest: ‘Langtang (Station ID 1000) is the highest elevation station among the six stations but has the fewest number of years of temperature observation…’
P5
L2: suggest revising ‘All the available’ to ‘All available’
L3: suggest: ‘Outliers were identified and removed…’
L5: revise to ‘assumption were applied’
L7: stations (should be plural)
L8-14: Did you use ‘Statgraphics’ to do this procedure? This section is a bit unclear.
P6
L9: suggest revising ‘were slightly less than’ to ;was slightly lower than’
P7
L1: either method (singular) and ‘is’ (singular) or ‘methods’ and ‘are’
L7-9: lacking a bit of clarity. Are all stations used in the rest of your analysis but in this case (for the change-point analysis) you are using a single station and you chose the one with the longest record? Are results from this one station then applied to the other 5 stations? How did you use the information from this analysis to inform the rest of your analysis? i.e. did you then compute trends before and after the identified change-points?
L13: ‘the Mann-Kendall’
L15-16: a bit unclear to me. How were these breaks identified at most of the stations?
L22: suggest revising ‘should be’ to ‘are’
L23: suggest revising ‘associated to’ to ‘associated with’
P8
L18:do you mean NOAA (instead of Noah?)
L19: ‘were used’
L20-30: references required. See general comments.
L26-27: unclear – Monthly average… were further averaged to obtain monthly averages’ Do you that you took an average of all stations? Please clarify.
L37-47: language could be more concise and direct. This would likely address general comment concerning time periods and comparisons.
L43-45: why only 2015? (see general comments)
P9
L3: ‘EMD methods were used to detect temperature trends at six’
L15: ‘mean annual temperature anomalies’
L17: unclear – ‘few extreme rise and fall are recorded’ Do you mean that there were few anomalously high or low temperatures?
L27: How does your study differed from Shrestha et al? Same or different stations? Same or different time period?
L30: at the 96%
L31: ‘that starts in’
P11 Table 3 – maybe period 1 and period 2 because it’s not really the first half and second half?
P11
L2: ‘significant’ – do you mean ‘statistically significant?’
L1-2: are those values the average of all 6 stations? A bit unclear.
L2-3: language a bit unclear. Suggest revision along the lines of: ‘For the period 1970-2015 all stations except Chapkot had statistically significant trends’ (or something similar; more clear language in remainder of paragraph might also help)
L8: why all of a sudden is 1993-2015 being used? Where did this come from. Please clarify.
L11-13: these two sentences seem to contradict each other. Or maybe I am missing something?
P12
L6: excellent to compare your results with other studies and to place your results in this context but sometimes it is unclear if or how your study is different from these existing studies. If your work is unique and adds something special I want to know that.
L10 – Heading – perhaps monthly and seasonal trends?
L11-12: suggested revision ‘…at each station (Figure 8) is lowest in January and highest during June-August.’
L13: perhaps say ‘the lowest’ instead of ‘minimum’ because using minimum can make a reader think you computed the average, maximum, and minimum. Similar confusion occurs elsewhere [e.g. next line with ‘maximum’]
L18: should be ‘months of April and December’ (months plural)
L25: suggest ‘The calendar year’ instead of ‘A year’
L26-27: ‘to assess the seasonal temperature trend’
L33: ass commas – ‘per year, respectively, and those’
L33: statistically significant at what confidence level?
L34-35: ‘at the basin and regional scales’
P14
L5: ‘The lapse rates were calculated’ (lapse rates plural)
P15
L6: suggest revising to: ‘The largest (…) and least negative temperature gradient…’.
L7-8: suggest: ‘…whereas … had the same lapse rate (…)’
L12: perhaps ‘The more gradual’ instead of ‘the very less steep’
L13: ‘are due to the strong’
L18 – an introductory sentence to this section (section 4.5) would be helpful.
L18-19: suggest revision to something along the lines of: ‘There is little difference in the mean monthly temperatures from the modeled temperature datasets (GLDAS 2.0, CHELSA 1.2) at…’
L20-21: A bit confusing here between time periods and datasets. This relates to general comment about this comparison. It’s not clear if you compared WorldClim data with in situ data for two separate time periods (1970-2000, 1979-2013) or if one time period was used for one product and another time period for another product.
L22: suggest ‘is expected’ or ‘is not surprising’ instead of ‘is obvious’
L29: It is the mean monthly LST temperature (not the mean monthly temperature from LST) MODIS is giving you skin temperature, your in-situ data is air temperature.
P17
L1-14: series of disparate sentences should be combined into a single paragraph.
L1-14: what is meant by ‘satisfactory correlation’?
L14: This line is important. Perhaps a few words to elaborate on the importance of this statement?
P19
L7: suggest ‘to identify’ instead of ‘to find out’
L8: what is meant by an ‘acceptable’ correlation?
L18: is this the average of all 6 stations?
L21-22: see general comment about lapse rate.
L23: although I don’t disagree here I am not sure where this conclusion came from in relation to the results presented. Perhaps it fits better as a general comment or statement? Perhaps I missed something?
Author Response
General comments: Introduction: Comment: very good review of existing work but I found I got lost in the details. Instead of listing each study separately is it possible to synthesize the results more generally? Or perhaps include a paragraph break to more clearly demonstrate summary of in-situ vs model-based studies? Methods: 3.3 Gap filling Please explain or elaborate on gap filling process and use of stations. In Section 3.1 you describe how stations were selected that represent different physiographic zones and climate ranges. Given this statement how would these differences affect the gap-filling process? How are these differences addressed in the gap-filling process, if at all? What would be the impact on the ‘filled’ data? 3.4-Trend break detection Longest record used for the EMD analysis. This makes sense but you need to state if and how the results for that station might be applicable to the other stations in your region. You specified that your stations cover different physiographic and elevation zones. Are the results obtained in the EMD analysis more a reflection of the broader climate and less so of the micro/local climate? If this is the case then you could assume what is happening at Nuwakot is probably also happening at the other stations. This is just something to consider. Other Was any attention paid to autocorrelation when conducting your trend analysis? If so how? If not, perhaps something to consider. Data MODIS – include additional references, see: Wan Z 2013Collection 6. MODIS Land Surface Temperature Products. Users’ Guide. [and references therein] Given the climate of the region, the absence of data when clouds are present is might be an important factor affecting the usefulness of optical satellite data. MODIS data used here are averages of all available cloud-free scenes within an 8-day period. MOD11A2 values are produced when there is a minimum of one cloud-free scene within a given 8 day period. This means that variability within that 8 day period can introduce variability in the MODIS record that is not physically real. In areas affected by cloud cover this could be a significant issue. The number of days used for each retrieval is provided in the QA layers. 3.6 – please include location and date of data access of data products either here or in the references. Add reference to MODIS data product (example is from LP DAAC): Wan, Z., S. Hook, G. Hulley. MOD11A2 MODIS/Terra Land Surface Temperature/Emissivity 8-Day L3 Global 1km SIN Grid V006. 2015, distributed by NASA EOSDIS Land Processes DAAC, https://doi.org/10.5067/MODIS/MOD11A2.006 4.4 - Temperature gradients The analysis of ‘lapse rate’ isn’t too convincing as the stations are in different physiographic regions. What is really being calculated is the line between temperature and elevation from your sample of stations. Lapse rate is usually at a given location. This is probably more a matter of terminology? However, in looking at the plot (figure 11) four of the stations are around 1000 m a.s.l. so it is clustered to the left. What I find more interesting than the lapse rate is that the seasonal difference in temperature is quite small at the high elevation station compared to the other 5 (I can’t tell if the mid-elevation station has a larger spread than the low elevation stations). There is something interesting here but I don’t know if it comes across the way it is described. 4.5 – Comparison of station data with freely available global climate datasets Data availability did not permit comparison with in-situ data and derived data over a common time period. Please state this and tell the reader more clearly how you compared the datasets. i.e. for each derived data product, in situ data for the corresponding period of record was compared. I am curious as to why only 2015 was used for MOD11A2. Data for this product are available beginning in 2000. Given that you are comparing different time periods, and your trend analysis showed changes in temperature, would you expect to have similar average temperature values? Please provide some mention of this caveat to ensure your comparisons are properly interpreted by the reader. Specific comments Line numbers refer to lines of text on each page. Figures, Tables and captions not included in numbering. Overall, suggest yr-1 instead of per year, and decade-1 instead of per decade P1 – Abstract L2: downstream of what? L3: glacier ice L5: Near surface (2m) temperature data? Specify somewhere near the beginning that you are looking at air temperature, except of course where you are looking at skin temperature. L6: m a.s.l. L8: remove ‘the’ from ‘the missing data’ L9: add ‘a’ to ‘a Mann-Kendall’ L9: revise ‘for testing the’ to ‘to test the’ L10: ‘trend’ should be ‘trends’ plural L12: by ‘mean increasing trend’ do you mean the mean of all six stations? Please clarify P1 introduction L3: remove ‘the’ from ‘the global’ L4: ‘are significantly rising since last decade’. Suggest revising to: ‘have been rising significantly over the last decade’ L6: change by 0.12°C per decade to ‘to 0.12 decade-1’ L7: change ‘the rate of increase of temperature are different’ to ‘the rate of temperature increase is’ L8-9: note – mountain glacier melt contributes to the eustatic component of SLR. [as opposed to thermal expansion] P2 L1: revise ‘This warming are’ to ‘this warming is’ L6-7: you mention ‘several studies’ but only once citation. Does this citation discuss multiple studies or is more than one reference necessary. COLLEEN TO CHECK L17: revise ‘by 0.03°C’ to ‘of 0.03°C L18: and ‘by 0.38°c’ to ‘of 0.38°C L19: suggest revising to something along the lines of: ‘found that increases in maximum temperature (0.05 °C yr-1) were larger than those of the minimum temperature… with the largest change observed during the pre-monsoon season…’ L22: revise ‘in average temperature at a rate of’ to ‘in average temperature of’ L24: revise ‘The model’ to ‘Model’ L24: ‘the Koshi River basin’ (add ‘the’) L25: suggest revising ‘due to rise in temperature’ to ‘due to increasing temperatures’ L27: ‘the Koshi trans-boundary…’ L45 (study area): revise ‘between latitude….85.83°E’ to ‘between 25.49°N-29.28°N and 85.02°E-85.83°E’ L47: ‘at the southern edge’ L48: km² instead of sq. km L48-49: ‘It encompasses a total area of…, and area within Nepal is about…’ to ‘It encompasses a total area of 46,300 km², 32,104 km² of which is in Nepal’ P3 L3: ‘The upper part of this basin is covered with snow’ (add ‘is’) L7-8: suggest revising to ‘large variations in climate over the basin’ L10: suggest revising to ‘in the monsoon season’ L15 (methods, data availability): specify/clarify if it is daily air temperature. L16 (methods, data availability): ‘were selected (Table 1) from different …’ to ‘were selected (Table 1), each representing different…’ L19: ‘and have data gaps longer than 5 years’. Unclear if you mean none of the stations have data gaps >5yrs or if they all have data gaps longer than 5yrs. L19: suggest revising to: ‘Langtang station has the shortest temperature record’ L21: are not uniform in what sense? Slightly ambiguous. P4 L2: suggest: ‘Langtang (Station ID 1000) is the highest elevation station among the six stations but has the fewest number of years of temperature observation…’ P5 L2: suggest revising ‘All the available’ to ‘All available’ L3: suggest: ‘Outliers were identified and removed…’ L5: revise to ‘assumption were applied’ L7: stations (should be plural) L8-14: Did you use ‘Statgraphics’ to do this procedure? This section is a bit unclear. P6 L9: suggest revising ‘were slightly less than’ to ;was slightly lower than’ P7 L1: either method (singular) and ‘is’ (singular) or ‘methods’ and ‘are’ L7-9: lacking a bit of clarity. Are all stations used in the rest of your analysis but in this case (for the change-point analysis) you are using a single station and you chose the one with the longest record? Are results from this one station then applied to the other 5 stations? How did you use the information from this analysis to inform the rest of your analysis? i.e. did you then compute trends before and after the identified change-points? L13: ‘the Mann-Kendall’ L15-16: a bit unclear to me. How were these breaks identified at most of the stations? L22: suggest revising ‘should be’ to ‘are’ L23: suggest revising ‘associated to’ to ‘associated with’ P8 L18:do you mean NOAA (instead of Noah?) L19: ‘were used’ L20-30: references required. See general comments. L26-27: unclear – Monthly average… were further averaged to obtain monthly averages’ Do you that you took an average of all stations? Please clarify. L37-47: language could be more concise and direct. This would likely address general comment concerning time periods and comparisons. L43-45: why only 2015? (see general comments) P9 L3: ‘EMD methods were used to detect temperature trends at six’ L15: ‘mean annual temperature anomalies’ L17: unclear – ‘few extreme rise and fall are recorded’ Do you mean that there were few anomalously high or low temperatures? L27: How does your study differed from Shrestha et al? Same or different stations? Same or different time period? L30: at the 96% L31: ‘that starts in’ P11 Table 3 – maybe period 1 and period 2 because it’s not really the first half and second half? P11 L2: ‘significant’ – do you mean ‘statistically significant?’ L1-2: are those values the average of all 6 stations? A bit unclear. L2-3: language a bit unclear. Suggest revision along the lines of: ‘For the period 1970-2015 all stations except Chapkot had statistically significant trends’ (or something similar; more clear language in remainder of paragraph might also help) L8: why all of a sudden is 1993-2015 being used? Where did this come from. Please clarify. L11-13: these two sentences seem to contradict each other. Or maybe I am missing something? P12 L6: excellent to compare your results with other studies and to place your results in this context but sometimes it is unclear if or how your study is different from these existing studies. If your work is unique and adds something special I want to know that. L10 – Heading – perhaps monthly and seasonal trends? L11-12: suggested revision ‘…at each station (Figure 8) is lowest in January and highest during June-August.’ L13: perhaps say ‘the lowest’ instead of ‘minimum’ because using minimum can make a reader think you computed the average, maximum, and minimum. Similar confusion occurs elsewhere [e.g. next line with ‘maximum’] L18: should be ‘months of April and December’ (months plural) L25: suggest ‘The calendar year’ instead of ‘A year’ L26-27: ‘to assess the seasonal temperature trend’ L33: ass commas – ‘per year, respectively, and those’ L33: statistically significant at what confidence level? L34-35: ‘at the basin and regional scales’ P14 L5: ‘The lapse rates were calculated’ (lapse rates plural) P15 L6: suggest revising to: ‘The largest (…) and least negative temperature gradient…’. L7-8: suggest: ‘…whereas … had the same lapse rate (…)’ L12: perhaps ‘The more gradual’ instead of ‘the very less steep’ L13: ‘are due to the strong’ L18 – an introductory sentence to this section (section 4.5) would be helpful. L18-19: suggest revision to something along the lines of: ‘There is little difference in the mean monthly temperatures from the modeled temperature datasets (GLDAS 2.0, CHELSA 1.2) at…’ There is little difference in the mean monthly temperatures from the modeled temperature datasets (GLDAS 2.0, CHELSA 1.2) at Pokhara and Chapkot. L20-21: A bit confusing here between time periods and datasets. This relates to general comment about this comparison. It’s not clear if you compared WorldClim data with in situ data for two separate time periods (1970-2000, 1979-2013) or if one time period was used for one product and another time period for another product. L22: suggest ‘is expected’ or ‘is not surprising’ instead of ‘is obvious’ L29: It is the mean monthly LST temperature (not the mean monthly temperature from LST) MODIS is giving you skin temperature, your in-situ data is air temperature. P17 L1-14: series of disparate sentences should be combined into a single paragraph. L1-14: what is meant by ‘satisfactory correlation’? L14: This line is important. Perhaps a few words to elaborate on the importance of this statement? P19 L7: suggest ‘to identify’ instead of ‘to find out’ L8: what is meant by an ‘acceptable’ correlation? L18: is this the average of all 6 stations? L21-22: see general comment about lapse rate. L23: although I don’t disagree here I am not sure where this conclusion came from in relation to the results presented. Perhaps it fits better as a general comment or statement? Perhaps I missed something? Dear Reviewer, We are very much thankful to our reviewer who provided very insightful comments and suggestion to our manuscript. Your suggestions and comments helped to increase the quality of the manuscript. We have tried our best to address all suggestions and comments suggested by you. The responses to each comment are provided below. Introduction: We have tried to improve the language clarity to make it easy to understand in our revised manuscript. 3.3. Gap Filling: We have tried to improve the language clarity to make it easy to understand in our revised manuscript. The following part is added in the manuscript. "Although the selected stations are in the different climatic zone as stated above, while developing the regression models to fill data gaps, independent variables selection are solely based on its statistically significant correlation with independent variables. Therefore the impact of the station in different physio-graphic and elevation zones assumed to be minimal. 3.4-Trend break detection: Thank you for your comments and feel sorry for not making clear while writing the manuscript. Actually, longest time-series is used only to show into the manuscript but all station data were analyzed by using EMD methods to detect trends. Now the manuscript is revised accordingly. Other: Yes, we have calculated auto-correlation for each station data-sets to check its seasonality. We found that most of the station has auto-correlation (1-day lag) more than 0.8 and concluded that the data are seasonal. Data: We added the reference in the revised manuscript. One limitation while comparison with MODIS data set could be the absence of data when clouds are present. This is quite evident as clouds are persistent during the monsoon period in the area where weather stations are located. This also highlights an important factor related to the applicability of optical satellite data. As MODIS data used in this study are averages of all available cloud-free scenes within an 8-day period, variability within 8 day period can introduce significant variability in the MODIS data set which may not represent the actual physical condition. This can always remain an issue while using MODIS data set for comparison with station data which represent actual ground conditions. As we have compared with three more datasets, use of MODIS datasets just one more. The correlation we obtained is quite high, which is acceptable. 3.6: We have added this reference in the revised manuscript. 4.4: Temperature gradient (lapse rate) is the rate at which temperature changes with altitude and it needs at least two stations at a different altitude. We estimated this rate to understand the decrease of temperature with elevation in our study site and it can help to predict the temperature for the sites, where stations are not available. Usually, two stations at different altitude are enough to estimate the temperature gradient if local weather condition does not change significantly with different valleys and mountains. High mountain topography and deep valleys within our study site need several stations at different altitudes and different valleys to accurately estimate the lapse rate. We were limited by the data availability and used the available dataset to estimate the lapse rate. Yes, you are right that temperature varies less in the higher mountain and greater in lower altitudes. 4.5 - Comparison of station data with freely available global climate datasets: One important reason to compare the station dataset with the freely available dataset. There was a big data gap in our dataset and it is also necessary to compare with other datasets to check whether our gap filled dataset match with modeled dataset or not. We assumed the comparison with one-year data is enough to check the reliability of station data. Therefore, we chose only one year for MOD11A2 dataset. P1-Abstract L2: We considered the upstream region from where river originated (Water source area) and downstream refers to the lower part of the basin, where people have easy access to use the water. L3: Modification was done in the revised manuscript L5: Yes, we have used the near surface temperature data (⁓2 m). L6: Modification was done in the revised manuscript L8: It is modified in the revised manuscript L9: Modification was done in the revised manuscript L9: Modification was done in the revised manuscript L10: Modification was done in the revised manuscript L12: Yes, it was mean of six stations P1 Introduction L3: Modification was done in the revised manuscript L4: Thank you for giving nice suggestion and it modified in the revised manuscript L6: We have modified this and all similar cases in the whole manuscript L7: Modification was done in the revised manuscript L8-9: Here, we removed the word “mountain” to include all the ice reserves of the world which contribute to SLR. P2 L1: Modification was done in the revised manuscript L6-7: We have included more citation in revised in version L17: we have revised ‘by 0.03°C’ to ‘of 0.03°C L18: We made correction in revised in manuscript L19: Thank you for advising revision of this sentence, we have revised it in manuscript L22: We made correction in the revised in manuscript L24: We made correction in the revised in manuscript L24: We made correction in the revised in manuscript L25: We made correction in the revised in manuscript L27: We made correction in the revised in manuscript L45: We made correction in the revised in manuscript L47: We made correction in the revised in manuscript L48: We made correction in the revised in manuscript L48-49: We made correction in the revised in manuscript P3 L3: Sorry for typo error and modification is made in the revised manuscript L7-8: We made correction in the revised in manuscript L10: We made correction in the revised in manuscript L15: Yes, we obtained the daily temperature data from the Department of Hydrology and meteorology, Government of Nepal. L16: We made correction in the revised in manuscript L19: We have modified the sentence as follows: None of the stations have data for the entire period and some of them have data gaps of more than 5 years. L19: We made correction in the revised in manuscript L21: Thank you for your suggestion, and we have done modification in the revised manuscript. For example Data available from all these stations are from the different time period. P4 L2: Modification is made in the revised manuscript P5 L2: We made correction in the revised in manuscript L3: Modification is made in the revised manuscript L5: We made correction in the revised in manuscript L7: We made correction in the revised in manuscript L8-14: We have made modification here that we have discarded the term “stagraphics”. This term was used for stepwise regression analysis. P6 L9: We made correction in the revised in manuscript P7 L1: We made correction in the revised in manuscript L7-9: Sorry for the misunderstanding, we have applied the same techniques to all stations to detect the trend break and we have modified our manuscript as per. L13: We made correction in the revised in manuscript L15-16: We have applied an empirical mode decomposition methods for detecting trends by decomposing data into local oscillatory components i.e. intrinsic mode functions (IMFs). L22: We made correction in the revised in manuscript L23: We made correction in the revised in manuscript P8 L18: It Is Noah, for your reference here is the link of the dataset: https://disc.gsfc.nasa.gov/datasets/GLDAS_NOAH025_M_V2.0/summary?keywords=GLDAS L19: Correction is done in the revised in manuscript L20-30: Reference is provided modified in the revised manuscript L26-27: We averaged monthly average temperature for day time and night time to get monthly mean temperature and it was done for all stations. L37-47: We are very grateful to the reviewer for the suggestion. However, since we are dealing with several stations and temperature data sets from multiple sources and for different time periods for our analysis, the concise explanation could increase ambiguity and therefore, lengthy explanations were deliberately provided to reduce the ambiguity and increase the clarity of our manuscript. L43-45: The main purpose of using modeled data is to check the validity of gap-filled datasets. We have compared the WroldClim, GLDAS and CHELSA dataset from 1970-2000 and 1979-2013, respectively for this purpose. MODIS data was just one additional dataset and we assumed one-year data is enough for this case. P9 L3: Correction is done in the revised in manuscript L15: We have modified in revised manuscript per your suggestion. L17: Yes, we observed few extreme years that have temperature more than 2 degree Celsius higher than the long-term average temperature. This might be related to drought or dry year. But it needs further investigation to check i.e. precipitation. L27: He has done his study in whole Nepal by selecting the 41 stations and his study period was from 1971-1994. Only one station is common in his study and our study. L30: We have modified in revised manuscript per your suggestion. L31: Correction is done in the revised in manuscript P11: We have modified in revised manuscript per your suggestion. P11 L2: Correction is done in the revised in manuscript L1-2: Yes, those are the average values for all studied stations for two different periods. L2-3: We have tried to make it much easy to understand, we hope it is understandable now. L8: We have chosen Langtang station as this is located at the highest elevation (3800m) to understand the warming rate at higher elevation and data for this station was available only after 1993. L11-13: We have checked the warming rates at different elevations for the different periods, which does not show a clear relationship for all period. However, when we compare just two stations i.e. lowest (Chapkot) and highest (Langatng), it shows the higher warming rate at a higher elevation than lower elevation. P12 L6: Our study was focused on the Narayani River basin, where previous studies are not documented. Also, we have used the dataset from much earlier than others, who have conducted their studies either in regional scale or in the different river basin. Further, we have check the validity of our dataset by comparing with a different freely available dataset. L10: Correction is done in the revised in manuscript We have no calculated the monthly temperature trend here. We just estimated the lowest mean temperature month and highest mean monthly temperature. L11-12: Modified in the revised manuscript L13: Thank you for your suggestion, we have changed it in the revised manuscript L18: Correction is done in the revised in manuscript L25: Correction is done in the revised in manuscript L26-27: Correction is done in the revised in manuscript L33: Modified in the revised manuscript L33: It was 95% and revised as per suggestion L34-35: Correction is done in the revised in manuscript P14 L5: Correction is done in the revised in manuscript P15 L6: We have modified in revised manuscript per your suggestion. L7-8: Correction is done in the revised in manuscript L12: Correction is done in the revised in manuscript L13: We have modified in revised manuscript per your suggestion. L18: Thank you very much for the suggestion. We have added the following introductory sentence. "Mean monthly temperatures from different data sets were observed for all six stations". L18-19: We have revised in the revised manuscript as follows: "There is little difference in the mean monthly temperatures from the modeled temperature datasets (GLDAS 2.0, CHELSA 1.2) at Pokhara and Chapkot". L20-21: We are sorry for the typological error. We have made modification in the revised manuscript as follows: There is no significant difference between mean monthly temperature values from WorldClim 2.0 (1970–2000) and station data (1970–2000). L22: We have modified in the revised manuscript per your suggestion. L29: We understand this and we are very grateful to the reviewer for pointing this out. By ‘mean monthly temperature from LST’, we mean the value obtained for mean monthly LST temperature obtained from MODIS LST product. P17 L1-14: We tried removing the disparity but since we are only dealing with results of comparisons of output from different climate data sets and for different stations, these sentences had to seem desperate. L1-14: We have now replaced the term with an acceptable correlation. Thank you for your kind suggestion to improvise our manuscript. L14: We have added the following information at this line. The consistency between gap-filled data set and freely available climate data set justifies the procedure for gap filling which is usually necessary for the mountain reason where malfunctioning of weather stations is usually frequent. This also shows that freely available climate data sets are very good alternatives for remote areas which are otherwise lacking data set from well-monitored weather stations. P19 L7: We have modified in revised manuscript per your suggestion. L8: We observed the very high correlation (>0.90) between observed temperature data and freely available dataset for most of the cases except for Chapkot, Nuwakot, and Langtang with GLDAS dataset. While all other dataset showed high correlation and we assumed that our gap filling techniques were good enough to fill the gap in the dataset. Also, from this correlation, it is acceptable that different freely available dataset can be used where data are not available. L18: Yes it was mean for all station L21-22: We have responded it in general comment L23: Thanks again for this critical comments, we made the modification of this point per your suggestion in the revised manuscript. We would like to express our sincere thanks to the reviewer, who provided insightful suggestions and comments. We hope, the manuscript is significantly improved after revision.Round 2
Reviewer 2 Report
I appreciated the authors’ effort to address the editorial comments and suggestions I made in the previous revision. The authors answered point by point and satisfactorily to my observations, but in some cases they just provided an explanation within their response letter and not within the manuscript. Overall, I think the authors still need to better clarify some points within the manuscript, which will be then ready to be published.
Below I have listed minor comments and suggestions that, in my opinion, the authors should address.
Most of the stations showed the cooling trend before the 1970s, however, station i.e Langtang shows a cooling trend till 1993. The original dataset of Langtang was ranging from 1988-2008 and we filled the data before and after this period using the simple and multiple linear regression methods. The earlier period was dominated by the cooling trend and it might have influenced the data filling method where there was a continuous gap before 1988 (1993). And, a minor variation of trend break year might be influenced by the variation local weather condition. Usually, we understood that carbon emission before the 1970s was limited and it was increased rapidly after this which led to rapid global warming. This effect is reflected in our results.
I totally agree with your explanation, and I strongly suggest you to add few lines in the manuscript to discuss this possible issue. I understand that Langtang station is the highest one and that it is possible that this implies a skip of trend break, but it seems to me that the difference with the other stations is that important. Therefore, you should underline that the lack of data before 1988 could affect the analysis that you perform on this station even after the filling procedure that you performed.
We have incorporated the river network of the Narayani River basin in the modified version of the manuscript.
Thank you.
Thank you very much for your comments, we have revised the manuscript per your suggestions. Similarly, we have added the sentence "Similar observed skewness from all station indicates similar seasonality in all stations" in the revised manuscript We have adjusted the figure size, their fonts, and dimensions for all figure in our revised manuscript.
Thank you.
We feel sorry for not making a clear citation. Now Xu et al. (2017) is added to make it clear. IMF6 was the final output (IMF), from which we can find out the trend break (either monotonically downward or monotonically upward). The other IMFs were intermediate output and from which we cannot identify the just two decompositions. However, those results are useful when we were interested in a short-term anomaly of climate in response to different climatic events.
I understand that other IMF’s are intermediate output, but I think that you don’t need to show them if they don’t provide any useful information for your analysis. At least, specify what you stated here, underlining that they can be useful for short term analysis.
Thank you for your critical reviewing the manuscript. You are right and your suggestion is appreciated. Most of the stations showed the cooling trend before the 1970s, however, station i.e Langtang shows a cooling trend till 1993. The original dataset of Langtang was ranging from 1988-2008 and we filled the data before and after this period using the simple and multiple linear regression methods. The earlier period was dominated by the cooling trend and it might have influenced the data filling method where there was a continuous gap before 1988 (1993).
As I previously stated, I strongly think that you should discuss this point in the text.
Thank you for your comments and suggestion. Actually, a significant warming trend in the winter is because of decreasing the number of winter days which is also indicated by Devkota (2014) in their study. And a change in temperature during monsoon, pre-monsoon and postmonsoon are could be because of changes in monsoon periods in the regions. Reference to justify this statement is now added to the revised manuscript. Now the explanation with references is added into the main text.
Thank you.
We have tried to explain it in terms of wind circulation in a different direction during different season and weather patterns.
Thank you
We are very grateful to the reviewer for these insightful suggestions. The intention behind displaying MAAT as the last point of the plot was to provide with a graphical representation for comparison of MAAT with mean monthly temperature values for 12 months. The intention behind not showing as a separate point was to display the sharp rise of the MAAT temperature values after December. This was done to emphasize the temperature difference between the maximum and minimum monthly temperature values and also to show that the MAAT values generally lie somewhere in between the maximum and minimum monthly temperature values, generally in the range of temperatures values observed for pre-monsoon (March-April) or post-monsoon (October) months. We totally agree with the reviewer that the representation might be misleading. Therefore, the labels in X-axis have been clearly explained in the figure captions. Both axes labels have been explained in the figure captions. We tried adding axes labels in the figure itself but it seemed redundant and unnecessary. We hope that the reviewer will kindly consider this situation.
I appreciate your explanation and I understand your intent, but still I feel that adding MAAT as the last point of the plot isn’t correct and it does not highlight the fact that MAAT lies between the maximum and minimum monthly temperature. Why don’t you consider the idea of showing the MAAT as a horizontal line in the plot? That would better highlight the comparison between MAAT and the monthly temperature of each month, and in my opinion it would be clearer.
We are again very grateful to the reviewer for this insightful suggestion. The basic intention behind considering MAAT values when fitting the trend lines shown in Figures 14-15-16-17 was to display that MAAT values lie in close agreement with other temperature values in the scatter plot. This would be strong evidence in support of our analysis. The next reason behind considering MAAT values when fitting the trend lines was to increase the number of points (although only by one point) for comparisons made for temperature data from MODIS, CHELSA and WorldClim 2.0 as these temperature data sets have only 12 data points in the scatter plot for comparison. However, as the reviewer has already mentioned, adding these points would not change the results of the analysis but that is one another evidence in support of our analysis that MAAT values are in close agreement with another set of data points in the scatterplot. We hope that the reviewer will kindly consider this situation.
Thank you, I appreciate your explanation.

Author Response
I appreciated the authors’ effort to address the editorial comments and suggestions I made in the previous revision. The authors answered point by point and satisfactorily to my observations, but in some cases they just provided an explanation within their response letter and not within the manuscript. Overall, I think the authors still need to better clarify some points within the manuscript, which will be then ready to be published. Below I have listed minor comments and suggestions that, in my opinion, the authors should address. I totally agree with your explanation, and I strongly suggest you to add few lines in the manuscript to discuss this possible issue. I understand that Langtang station is the highest one and that it is possible that this implies a skip of trend break, but it seems to me that the difference with the other stations is that important. Therefore, you should underline that the lack of data before 1988 could affect the analysis that you perform on this station even after the filling procedure that you performed. We are very thankful to reviewer for providing in-depth comments and suggestion to our manuscript. We tried our best to address all the comments and suggestions and revised the manuscript accordingly. Thank you for providing insightful suggestions and now we have added the following information in the revised manuscript in the second paragraph of section 4.2. Trend break was observed in the 1970s for all stations i.e. in 1970 for Jomsom and Pokhara, in 1972 for Chapkot and Dhunibesi, in 1979 for Nuwakot, which is cross-validated with EMD results. However, the cooling trend for Langtang was detected in 1993, in which original dataset was ranging from 1988--2008 and data was extended for 1960--2015 by gap-filling techniques. The earlier period was dominated by the cooling trend in most of the stations and it might have influenced the data filling method and cooling trend for Langtang was extended up to 1993 from where continuous data was available. I understand that other IMF’s are intermediate output, but I think that you don’t need to show them if they don’t provide any useful information for your analysis. At least, specify what you stated here, underlining that they can be useful for short term analysis. We totally agree with your comments that IMF’s are only intermediate output. But we think each IMF described some of the data property. Now in the manuscript revised and now reads as follows in the trend break observation section: EMD methods were used to detect temperature trends in six stations within the Narayani River basin. Intrinsic mode functions (IMFs) 1 exhibit high frequency and can represent very short-term fluctuation (Figure~\ref{fig:fig_6}), IMFs 2--3 captures a small percentage of variance, IMFs 4--5 capture mid-term effects described by periodic cyclic variation~\cite{mahmoud_2009}. Finally, IMF 6, the residue component in EMD could represent the major trend~\cite{thapa_2015} of annual average temperature in the long term that may be related with the increase or decrease of observed temperature in Pokhara station. From Figure ~\ref{fig:fig_6}, high fluctuation during the periods between 1960 and 1970 is observed in IMF1. Higher fluctuation represents higher temperature variability during the period 1960 and 1970. Higher variability during that periods impacts mid periodic cycle, shown by IMF4 as similar results are also discussed by ~\cite{mahmoud_2009} and ~\cite{thapa_2015} in their research. The annual average temperature is in a decreasing trend till 1972 and it breaks there and again the trend is in increasing order and represents the major trend of temperature (Figure~\ref{fig:fig_6}). It may be related to rapid global warming after the 1970s with industrial evolution and increase of greenhouse gases (GHGs) and radiative forcings~\cite{ipcc_2014}. The small difference in the number of year of trend break might be attributed to the influence of local and regional climate As I previously stated, I strongly think that you should discuss this point in the text. This specific comment is related to the cooling trend of Lang tang station and we have addressed this comment in response 1. I appreciate your explanation and I understand your intent, but still I feel that adding MAAT as the last point of the plot isn’t correct and it does not highlight the fact that MAAT lies between the maximum and minimum monthly temperature. Why don’t you consider the idea of showing the MAAT as a horizontal line in the plot? That would better highlight the comparison between MAAT and the monthly temperature of each month, and in my opinion it would be clearer. We have included a new figure (figure 13) which shows the MAAT values for all station in comparison with modeled dataset and added the following text for explaining the figure 13 in the second paragraph of section 4.5. MAAT values were estimated using monthly temperature values for all six locations using Worldclim 2.0, CHELSA, station data set (1970-2000), station data set (1979-2013), station data set (2001-2010) and GLDAS 2.0 data set (2001-2010) (Figure~\ref{fig:fig_13}). It can be observed that MAAT value obtained from station data set (1970-2000), station data set (1979-2013) and station data set (2001-2010) are similar for all locations. Compared to GLDAS 2.0 and Chelsa, MAAT obtained using WorldClim 2.0 is closer to the values obtained from station data set for Dhunibesi, Jomsom, Chapkot and Langtang. Large variations in MAAT values from different data set is observed for Jomsom, Nuwakot, and Dhunibesi.Reviewer 3 Report
Response to Author Revisions
The authors have made an excellent attempt to address the minor editorial comments and suggestions. The authors answered concerns regarding the use of MODIS temperature data for 2015 only but did not provide any evidence to support their answers and no additional analysis was conducted. Given that the MODIS data are freely available I would have liked to have seen, at a minimum, the MODIS data used for 2013-2015 to enable cross-comparison with the GLDAS data which were used until 2013. The reasoning here is that the MODIS data are heavily influenced by cloud cover and the region in question is also affected by cloud cover. No analysis of the MODIS data was conducted to investigate variability in cloud-free days within 8 day periods and is likely beyond the scope of this study. I understand that MODIS was used as simply another dataset but the use of s single year that does not overlap with any of the modeled products when multiple years are available does not seem adequate.
I have provided additional minor comments regarding specific comments that required further clarification or modification. ‘A’ denotes author’s answer. ‘R’ denotes reviewer response.
A: 3.3. Gap Filling: We have tried to improve the language clarity to make it easy to understand in our revised manuscript. The following part is added in the manuscript. "Although the selected stations are in the different climatic zone as stated above, while developing the regression models to fill data gaps, independent variables selection are solely based on its statistically significant correlation with independent variables. Therefore the impact of the station in different physio-graphic and elevation zones assumed to be minimal.”
R: Thank you for this addition. For readability and clarity, please consider the following modifications to the added text. “Although the selected stations are in the different climatic zone as stated above, while when developing the regression models to fill data gaps, independent variables selection are was based solely based on its statistically significant correlations with independent variables. Therefore the impact of the station in different physio-graphic and elevation zones is assumed to be minimal.”
R: P5 L1-2, suggested revision: “All available datasets for the six stations from 1960-2015 were visually inspected using several time series plots and outliers +/- three standard deviations [32] were removed.”
A: 3.4-Trend break detection: Thank you for your comments and feel sorry for not making clear while writing the manuscript. Actually, longest time-series is used only to show into the manuscript but all station data were analyzed by using EMD methods to detect trends. Now the manuscript is revised accordingly.
R: Thank you for clarifying this. Please add text in the manuscript to tell the reader you are only showing one station. Example: “For brevity, only the longest time-series (Pokhara station) is presented here.” Suggest adding similar sentence after the line “In this study we have used…”
A: L3: Modification was done in the revised manuscript L5: Yes, we have used the near surface temperature data (⁓2 m).
R: Please state somewhere early in the manuscript that you are using 2m air temperatures. P3 Section 3.1 Data Availability, Line 1 would be a good location. “Daily 2 m air temperature data….”
A: P1 Introduction L8-9: Here, we removed the word “mountain” to include all the ice reserves of the world which contribute to SLR.
R: Suggest adding ‘eustatic’ before ‘SLR’.
R: P3 Section 3 heading. Suggest simply ‘Data’
R: P5 line 9-10. Suggest removing “(i.e. surrounding station) for model development.” And simply end the sentence at “independent variables.”.
A: Now P6 end-P7 L7-9: Sorry for the misunderstanding, we have applied the same techniques to all stations to detect the trend break and we have modified our manuscript as per.
R: Thank you for the clarification. Suggested revisions to the added text: “Although the selected stations are in different climatic zones, as stated above, when developing the regression models to fill gaps, independent variables selection was solely based on its statistically significant (p<0.5) correlations with independent variables so the impact of station in different physiography and elevation zones were assumed to be minimal.”
R: P8 Line 7. Missing corresponding end apostrophe to go with ‘Annual Mean…
A: L17: Yes, we observed few extreme years that have temperature more than 2 degree Celsius higher than the long-term average temperature. This might be related to drought or dry year. But it needs further investigation to check i.e. precipitation.
R: I feel the explanation provided here is more precise and more informative than that included in the manuscript. I suggest replacing existing text with something along then lines of: “Few extreme annual temperature anomalies (+/- 2°C relative to the long-term mean) were observed.”
A: P10 L6 [was P9 L31]: Correction is done in the revised in manuscript.
R: Additional correction: should read “show” not “shows”
A: P12 L10: We have no calculated the monthly temperature trend here. We just estimated the lowest mean temperature month and highest mean monthly temperature.
R: OK. But perhaps ‘seasonal trends’ is not the most appropriate heading for this particular text?
A: L8: We have chosen Langtang station as this is located at the highest elevation (3800m) to understand the warming rate at higher elevation and data for this station was available only after 1993.
R: Thank you for the clarification. Perhaps include in parentheses a reminder to the reader of this fact. E.g. “at Langtang station was observed from 1993 to 2015 (data only available from 1993 onward, see Figure 2) …”
A: L11-13: We have checked the warming rates at different elevations for the different periods, which does not show a clear relationship for all period. However, when we compare just two stations i.e. lowest (Chapkot) and highest (Langatng), it shows the higher warming rate at a higher elevation than lower elevation.
R: Thank you for the clarification. I feel your explanation is clearer than what is in the text. Perhaps include or integrate with what is in the manuscript?
Perhaps integrate existing text with text below?
We investigated temperature trends at different elevations for the different time periods. This analysis did not show a clear relationship for all time periods and stations analyzed. However, comparison of the highest (Langtang) and lowest (Chapkot) elevation stations revealed higher rates of warming at Chapkot compared to Langtang.
A: P12 L6: Our study was focused on the Narayani River basin, where previous studies are not documented. Also, we have used the dataset from much earlier than others, who have conducted their studies either in regional scale or in the different river basin. Further, we have check the validity of our dataset by comparing with a different freely available dataset.
R: Is there a way to incorporate the above text into your manuscript?
A: L18: Thank you very much for the suggestion. We have added the following introductory sentence. "Mean monthly temperatures from different data sets were observed for all six stations".
R: Good attempt. I feel my suggestion may have not been on point so I have suggested the following, or something similar to: “Comparison of the mean monthly temperature from modeled datasets shows little difference in mean monthly temperature at Pokhara and Chapkot (Figure 12).” And then remove the first sentence.
A: L1-14: We have now replaced the term with an acceptable correlation. [L1-14: what is meant by ‘satisfactory correlation’?]
R: I think what you really mean here is statistically significant correlation?
Author Response
The authors have made an excellent attempt to address the minor editorial comments and suggestions. The authors answered concerns regarding the use of MODIS temperature data for 2015 only but did not provide any evidence to support their answers and no additional analysis was conducted. Given that the MODIS data are freely available I would have liked to have seen, at a minimum, the MODIS data used for 2013-2015 to enable cross-comparison with the GLDAS data which were used until 2013. The reasoning here is that the MODIS data are heavily influenced by cloud cover and the region in question is also affected by cloud cover. No analysis of the MODIS data was conducted to investigate variability in cloud-free days within 8 day periods and is likely beyond the scope of this study. I understand that MODIS was used as simply another dataset but the use of s single year that does not overlap with any of the modeled products when multiple years are available does not seem adequate. Dear Reviewer, Thank you very much again for providing valuable suggestions to our manuscript. We are very grateful for providing in-depth suggestions. Your suggestions helped to increase the quality of the manuscript. We tried our best to address all suggestions and comments suggested by you. The responses to each comment are provided below. Thank you for appreciating our work. Regarding the use of MODIS, we have the following response. Comparison of MODIS with GLDAS 2.0 had already been carried out for the year 2015 in the manuscript. Nevertheless, we are very grateful to the reviewer for the insightful suggestion. Therefore, we have carried out a comparison of GLDAS 2.0 and station data with MODIS LST data for the additional year 2014 as well and new figure (Figure 14) is added in the revised manuscript for the year 2014. We did not find any dissimilarity between the data of 2015 and 2014 of the MODIS. Comparison of MODIS with WorldClim 2.0 and CHELSA is not possible as they exist for different time periods so the comparison would not make any sense. Furthermore, our study objective has remained to compare station data with remotely sensed data sets and modeled data sets to ascertain the feasibility of applicability of remotely sensed and modeled data sets in the absence of station data for remotely located study areas in Nepal. Therefore, comparison between remotely sensed and modeled data set is a relatively new area for research and also somewhat beyond the scope of our study. We hope to do a similar study that focuses on the comparison of several modeled datasets including MODIS and station dataset for the Himalayan region in the future. Thank you for this addition. For readability and clarity, please consider the following modifications to the added text. “Although the selected stations are in the different climatic zone as stated above, while when developing the regression models to fill data gaps, independent variables selection are was based solely based on its statistically significant correlations with independent variables. Therefore the impact of the station in different physio-graphic and elevation zones is assumed to be minimal.” We have made correction in the manuscript in section 3.3 (Page 5) per your suggestion as follows: Although the selected stations are in different climatic zones, when developing the regression models to fill gaps, independent variables selection was solely based on statistically significant (p<0.05) correlations with independent variables so the impact of physiography and elevation were assumed to be minimal. P5 L1-2, suggested revision: “All available datasets for the six stations from 1960-2015 were visually inspected using several time series plots and outliers +/- three standard deviations [32] were removed.” We incorporated your suggestion in the revised manuscript in the first paragraph of quality control and data fill section. Thank you for clarifying this. Please add text in the manuscript to tell the reader you are only showing one station. Example: “For brevity, only the longest time-series (Pokhara station) is presented here.” Suggest adding similar sentence after the line “In this study we have used…” We have made correction in the manuscript per your suggestion as follows in trend break detection methods section, page 5. In this study, we have used the EMD method for trend break detection in all six stations, however, the only result of the longest time-series (Pokhara station) is presented for brevity. Please state somewhere early in the manuscript that you are using 2m air temperatures. P3 Section 3.1 Data Availability, Line 1 would be a good location. “Daily 2 m air temperature data….” We have made correction in the manuscript per your suggestion and incorporated the information regarding 2 m air temperature. Suggest adding ‘eustatic’ before ‘SLR’. We have made correction in the manuscript per your suggestion. P5 line 9-10. Suggest removing “(i.e. surrounding station) for model development.” And simply end the sentence at “independent variables”. Correction is done in revised manuscript. “Independent variables for the model were selected based on the highest significant correlation coefficient with surrounding stations in the model development process to fill the gap. To improve the predictive power of the model, step-wise regression methods were used to select multiple independent variables” Thank you for the clarification. Suggested revisions to the added text: “Although the selected stations are in different climatic zones, as stated above, when developing the regression models to fill gaps, independent variables selection was solely based on its statistically significant (p<0.5) correlations with independent variables so the impact of station in different physiography and elevation zones were assumed to be minimal.” We have made correction in the manuscript per your suggestion in section 3.3. “Although the selected stations are in different climatic zones when developing the regression models to fill gaps, independent variables selection was solely based on statistically significant (p<0.05) correlations with independent variables so the impact of physiography and elevation were assumed to be minimal.” P8 Line 7. Missing corresponding end apostrophe to go with ‘Annual Mean… Correction is done in the revised manuscript in section 3.6 and page 7, which can be read as follows. “The average monthly temperature data for 1970--2000 and ‘Annual Mean Temperature or BIO1 grid' from WorldClim Version 2.0 were used in this study.” I feel the explanation provided here is more precise and more informative than that included in the manuscript. I suggest replacing existing text with something along then lines of: “Few extreme annual temperature anomalies (⁓+/- 2°C relative to the long-term mean) were observed.” We have made correction in the manuscript (Section 4.2, page 8) per your suggestion and revised as follows: Few extreme events of temperature anomalies (+/-2 °C) relative to the long-term mean were observed in the data and also show the spatial variability in temperature as stations are located in different physiographic zones. Additional correction: should read “show” not “shows” We have made correction in the manuscript per your suggestion. OK. But perhaps ‘seasonal trends’ is not the most appropriate heading for this particular text? Thank you for pointing out this. This section is for the seasonal trend which we have described in the second paragraph. The first paragraph provides some basic information about monthly temperature for different station. Thank you for the clarification. Perhaps include in parentheses a reminder to the reader of this fact. E.g. “at Langtang station was observed from 1993 to 2015 (data only available from 1993 onward, see Figure 2) …” The data availability for the Langtang station is from 1988 to 2008 is stated in the data availability section. We did a gap-filling and trend break for this station was detected in 1993 which is presented in table 3. After 1993, we observed a warming trend in this station, therefore we provided information about the warming trend for the Langtang station after 1993. Thank you for the clarification. I feel your explanation is clearer than what is in the text. Perhaps include or integrate with what is in the manuscript? Perhaps integrate existing text with text below? We investigated temperature trends at different elevations for the different time periods. This analysis did not show a clear relationship for all time periods and stations analyzed. However, comparison of the highest (Langtang) and lowest (Chapkot) elevation stations revealed higher rates of warming at Chapkot compared to Langtang. We have integrated your suggestion in the manuscript and want to make clear that the highest warming rate was observed in Langtang, not at Chapkot (Section 4.2 and page 10), which can be now read as follows. “We investigated temperature trends at different elevations for the different time periods. This analysis did not show a clear relationship for all time periods and stations analyzed. However, comparison of the highest (Langtang) and lowest (Chapkot) elevation stations for the period 1980--2015 revealed higher warming rate at Langtang (0.044°C year-1) compared to Chaptkot (0.013 °C year-1).” Is there a way to incorporate the above text into your manuscript? (P12 L6) We have tried to integrate the information in section “Historical Observed trend” i.e. Studies related to temperature trends in the Narayani River basin are not well documented. Here, we have used the datasets from much earlier than the previous study, which was conducted either in regional scale or different river basins. However, our results are similar to Shrestha~et~al. (1999), which was done for the whole of Nepal. P12, L18: Good attempt. I feel my suggestion may have not been on point so I have suggested the following, or something similar to: “Comparison of the mean monthly temperature from modeled datasets shows little difference in mean monthly temperature at Pokhara and Chapkot (Figure 12).” And then remove the first sentence. We have made modification in the manuscript per your suggestion. I think what you really mean here is statistically significant correlation? We found strong to a very strong correlation between modeled dataset and station data, therefore, here we have used the term acceptable correlation. The R2 values greater than 0.80 was assumed very strong correlation and R2 values between 0.60 and 0.80 were considered a strong correlation. And we have revised it as per in revised manuscript. We have not carried out the statistics test for this particular as we found very high correlation coefficient values, which is also depicted in all the figures.